# SymRTLO: Enhancing RTL Code Optimization with LLMs and Neuron-Inspired Symbolic Reasoning

**Yiting Wang**[1,*] **Wanghao Ye**[1,*] **Ping Guo**[2,*] **Yexiao He**[1,*] **Ziyao Wang**[1]
**Bowei Tian**[1] **Shwai He**[1] **Guoheng Sun**[1] **Zheyu Shen**[1] **Sihan Chen**[3]
**Ankur Srivastava**[1] **Qingfu Zhang**[2] **Gang Qu**[1] **Ang Li**[1,†]

[1]University of Maryland    [2]City University of Hong Kong    [3]University of Southern California

{ywang144, wy891, yexiaohe, angliece}@umd.edu

pingguo5-c@my.cityu.edu.hk

## Abstract

Optimizing Register Transfer Level (RTL) code is crucial for improving the efficiency and performance of digital circuits in the early stages of synthesis. Manual rewriting, guided by synthesis feedback, can yield high-quality results but is time-consuming and error-prone. Most existing compiler-based approaches have difficulty handling complex design constraints. Large Language Model (LLM)-based methods have emerged as a promising alternative to address these challenges. However, LLM-based approaches often face difficulties in ensuring alignment between the generated code and the provided prompts. This paper introduces SymRTLO, a neuron-symbolic framework that integrates LLMs with symbolic reasoning for the efficient and effective optimization of RTL code. Our method incorporates a retrieval-augmented system of optimization rules and Abstract Syntax Tree (AST)-based templates, enabling LLM-based rewriting that maintains syntactic correctness while minimizing undesired circuit behaviors. A symbolic module is proposed for analyzing and optimizing finite state machine (FSM) logic, allowing fine-grained state merging and partial specification handling beyond the scope of pattern-based compilers. Furthermore, an efficient verification pipeline, combining formal equivalence checks with test-driven validation, further reduces the complexity of verification. Experiments on the RTL-Rewriter benchmark with Synopsys Design Compiler and Yosys show that SymRTLO improves power, performance, and area (PPA) by up to 43.9%, 62.5%, and 51.1%, respectively, compared to the state-of-the-art methods. Our code is available at https://github.com/NellyW8/SymRTLO

## 1 Introduction

Register Transfer Level (RTL) optimization is a cornerstone of modern circuit design flows, serving as the foundation for achieving optimal Power, Performance, and Area (PPA). As the earliest phase in the hardware design lifecycle, RTL development provides engineers with the most significant degree of flexibility to explore design patterns, make architectural trade-offs, and influence the overall design quality [8]. Engineers use hardware description languages (HDLs) like Verilog to describe circuit functionality. At this stage, decisions made have far-reaching implications, as the quality of the RTL implementation directly impacts subsequent stages, including synthesis, placement, and routing [40]. A well-optimized RTL not only ensures better design outcomes but also prevents suboptimal designs from propagating through the flow, leading to significant inefficiencies and costly iterations [13, 49].

---

*Equal contribution

†Corresponding author

39th Conference on Neural Information Processing Systems (NeurIPS 2025).

Despite its importance, RTL optimization remains a challenging and labor-intensive task. Engineers must iteratively refine their designs through multiple rounds of synthesis and layout feedback to ensure functionality and meet stringent PPA targets. This process becomes increasingly cumbersome as design complexity grows, with synthesis times scaling disproportionately, often taking hours or even days for a single iteration [13]. Consequently, designers frequently face numerous synthesis cycles to evaluate trade-offs and reach acceptable results. While modern electronic design automation (EDA) tools provide compiler-based methods to aid optimization, these approaches are inherently limited [46]. They rely heavily on predefined heuristics, making them ill-suited for adapting to unconventional design patterns, complex constraints, or dynamic optimization scenarios. As a result, the RTL optimization process demands significant expertise and effort.

Recent advances in artificial intelligence, particularly the advent of large language models (LLMs), have introduced a new paradigm for automating and optimizing RTL code [23, 25, 5, 1, 37, 15, 46, 14, 42, 41, 16, 26]. Leveraging the powerful generative capabilities of LLMs, researchers have demonstrated their potential to rewrite and optimize Verilog code automatically [46]. However, existing LLM-based approaches face critical challenges that limit their effectiveness. First, these models often fail to align their generated outputs with specified optimization objectives. The inherent limitations in logical reasoning within LLMs can lead to deviations from intended transformations, resulting in suboptimal or incorrect outputs. Second, despite their potential for automating code generation, current methods still heavily rely on traditional synthesis feedback loops for optimization. This reliance results in the inefficiencies of the synthesis process, failing to address the core issue of long design cycles.

**Our Proposed Framework.**   To address the critical challenges in RTL optimization, we introduce SymRTLO, the first neuron-symbolic system that seamlessly integrates LLMs with symbolic reasoning to optimize RTL code. SymRTLO significantly reduces the reliance on repeated synthesis tool invocations and enhances the alignment of LLM-generated results with intended optimization rules.

Designing such a system introduces significant challenges, which we address through the integration of carefully designed modules. The first challenge lies in the *generalization* of optimization rules. Traditionally, optimization patterns are scattered across code samples, books, and informal notes, making it difficult for compiler-based methods to formalize or apply them effectively. SymRTLO tackles this by employing an LLM-based rule extraction system, combined with a retrieval-augmented generation (RAG) mechanism and a search engine. This ensures that optimization rules are not only generalized but also efficiently retrievable from a robust library built from diverse sources.

Another critical challenge is the *alignment* of LLM-generated RTL code with the intended transformations, as LLMs often struggle to produce outputs that strictly adhere to the specified optimization objective, leading to unreliable and unexplainable results. To ensure alignment, SymRTLO employs Abstract Syntax Tree (AST)-based templates, which guide the LLM to generate code that satisfies syntactic and semantic correctness. For complex control flows or edge cases that exceed the capabilities of AST templates, the framework utilizes a symbolic generation module, designed to handle such scenarios dynamically while maintaining optimization quality.

In addition to alignment, *conflicts* often arise when different design patterns are required to meet distinct PPA goals. To address this, SymRTLO introduces a goal-oriented approach, where each optimization rule is explicitly tied to its intended objective. This enables selective application based on user-defined optimization goals, efficiently balancing these conflicts to deliver optimized designs without disproportionately compromising PPA metrics.

*Verification* in traditional RTL workflows demands significant manual effort for test case development. To address this, SymRTLO integrates an automated test case generator, streamlining verification while ensuring functional correctness.

Our key contributions are summarized as follows:

- **LLM Symbolic Optimization:** SymRTLO, the first framework to combine LLM-based rewriting with symbolic reasoning for RTL optimization.

- **Data Path and Control Path Optimization:** SymRTLO addresses critical challenges in both traditional EDA compilers and purely LLM-based approaches, particularly by aligning generated code with FSM and data path algorithms, balancing conflicting optimization rules, and improving explainability.

- **PPA Improvements:** `SymRTLO` demonstrates its efficacy on industrial-scale and benchmark circuits, surpassing manual coding, classical compiler flows (*e.g.,* Synopsys DC Compiler), and state-of-the-art LLM-based methods, achieving up to 43.9%, 62.5%, and 51.1% improvements in power, delay, and area.

## 2 Background and Motivation

LLMs have emerged as powerful tools for RTL design automation, with various approaches being developed since 2023. As summarized in Table 1, these approaches fall into three primary categories: *RTL code generation* [23, 25, 5, 1], *debugging* [37, 15], and *optimization* [46]. This growing body of research demonstrates the significant potential of LLMs to improve the efficiency and effectiveness of EDA workflows. However, RTL code optimization has always been a significant challenge in RTL design, even for human experts, as it has the greatest impact on the performance of downstream tasks.

**Challenges in RTL Code Optimization with LLMs.** Aligning generated code with intended optimization goals is a major challenge in LLM-based RTL optimization. Due to inherent randomness, LLMs often produce incomplete, incorrect, or suboptimal results. For example, RTLRewriter [46] employs retrieval-augmented prompts and iterative synthesis-feedback loops to enhance functional correctness but still struggles with fundamental misalignment between generated code and optimization objectives. Additionally, the need for multiple synthesis rounds significantly increases optimization time as design complexity increases, limiting the scalability of current LLM-based methods for large industrial-scale designs.

**Current Approaches: Underutilization of Knowledge and Manual Verification.** Traditional RTL design optimization relies on established patterns such as subexpression elimination [31, 9], dead code elimination [20, 17], strength reduction [10], algebraic simplification [3, 4], Mux reduction [6, 43], and memory sharing [21, 27]. While these techniques are effective, these optimizations typically operate at the gate-level netlist, making the relationship between optimized output and original RTL code less transparent. Additionally, optimization patterns from design manuals and codebases remain underutilized due to the lack of a centralized repository, forcing engineers to rely on their expertise rather than automated tools. Furthermore, verification requires the creation of test benches and test cases manually, making the RTL design flow both time-consuming and error-prone.

Table 1: Comparative analysis of LLM-based methods for RTL design. ✔ indicates the presence of the feature, ✘ indicates absence of the feature, and – indicates non-applicable.

| Category | Method | Verification Capability | Rule -Based | Output Alignment | Conflict Resolution |
|---|---|---|---|---|---|
| Generation | ChipNeMo [23] | ✘ | ✘ | ✘ | – |
| | VeriGen [36] | ✘ | ✘ | ✘ | – |
| | VerilogEval [24] | ✘ | ✘ | ✘ | – |
| | RTLCoder [25] | ✘ | ✘ | ✘ | – |
| | ChipChat [1] | ✘ | ✘ | ✘ | – |
| | ChipGPT [5] | ✘ | ✘ | ✘ | ✘ |
| Debug | RTLFixer [37] | ✔ | ✔ | ✘ | ✘ |
| | LLM4SecHW [15] | ✘ | ✘ | ✘ | ✘ |
| Optimization | RTLRewriter [46] | ✔ | ✔ | ✘ | ✘ |
| | SymRTLO (**Ours**) | ✔ | ✔ | ✔ | ✔ |

**Optimization Conflicts and Limited Compiler Capabilities.** Existing compiler-based methods face additional challenges, particularly in managing **optimization goal conflicts** and handling complex patterns. For instance, optimizing for one metric, such as delay, often conflicts with another, such as power consumption. Striking the right balance between these competing objectives is crucial, especially as trade-offs between power and delay directly impact overall system performance. As shown in Table 2, each optimization method has its own specific goal, which often clashes with others. Compiler-based methods also lack the flexibility to adapt to such conflicts, limiting their effectiveness in optimizing designs with diverse and competing constraints.

Table 2: Conflicts Between Design Goals & Optimization Patterns.

| Design Pattern | Goal | Conflict Goal | Conflict Design Pattern |
|---|---|---|---|
| Pipelining | Low Timing | Low Area | Resource Sharing |
| Clock Gating | Low Power | Low Timing | Retiming |

Table 3: Performance Comparison Across Different Approaches.

| Approach | # States | Time (ns) | Power (mW) | Area ($\mu m^2$) |
|---|---|---|---|---|
| Baseline | 11 | 0.041 | 2.250 | 833.000 |
| GPT-O1 | 10 | 0.041 | 2.280 | 993.480 |
| Optimized | 4 | 0.025 | 1.170 | 403.920 |

**Advancing LLM-based RTL Optimization with Neuron-Symbolic Integration.** Recent research has shown a growing trend toward combining symbolic reasoning with LLM [45, 39, 7, 12, 11],

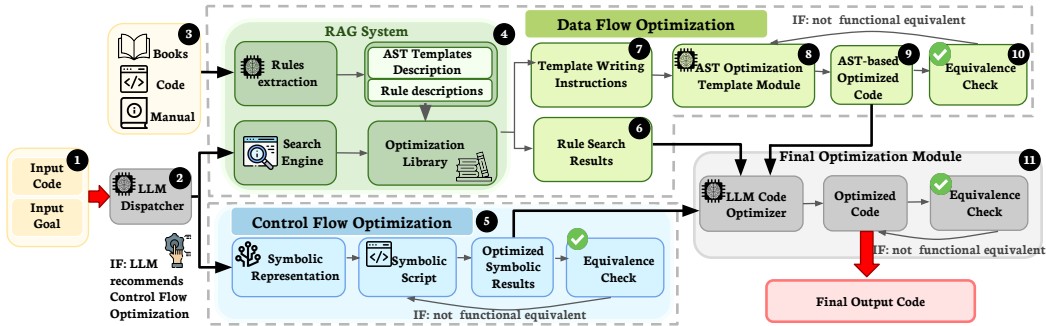

Figure 1: SymRTLO Architecture.

bringing new inspirations for more efficient and reliable LLM-based RTL code optimization with better prompt-code alignment. These integrated approaches have seen broad application across various fields, from automated theorem proof and knowledge representation to robotics and medical diagnostics, demonstrating how the combination of pattern recognition and generative capabilities of LLM with the interpretability and logical rigor of symbolic systems can significantly improve the alignment between LLM output and the given prompt.

**Motivation Experiments.** To highlight the limitations of current LLM-based RTL optimization, we conduct an experiment using state-of-the-art commercial LLM, GPT-O1, to optimize an 11-state FSM design. The goal was to minimize and merge unnecessary states to enhance PPA metrics. GPT-O1 receives a detailed state reduction algorithm to guide the optimization process. We compare its results with an optimized design that directly applied the state reduction algorithm. As shown in Table 3, GPT-O1 struggles to align its outputs with the algorithm, resulting in an under-optimized FSM with little state reduction and degradation in PPA. Although GPT-O1 achieves some state reduction, it fails to consider switching activity minimization and efficient state encoding, resulting in poor code choices that increase Hamming distances between frequently transitioning states and generate more complex combinational logic [48]. In contrast, algorithm-driven optimization achieves significantly better results, highlighting current LLM limitations in complex RTL optimization tasks.

## 3 Methodology

SymRTLO takes a Verilog RTL module as input and optimizes it for specific design goals, such as low power, high performance, or reduced area. As illustrated in Figure 1, the workflow begins by entering the RTL code and the user-specified optimization goal (①) into the **LLM Dispatcher** (②). This dispatcher analyzes the input circuit and determines the appropriate optimization path: either proceeding solely with **Data Flow Optimization** (§3.2) or incorporating **Control Flow Optimization** (§3.3) as well, depending on the characteristics of the design. For Data Flow Optimization, a search engine with a retrieval-augmented module extracts optimization rules and constructs AST-based templates. For Control Flow Optimization, an LLM-driven symbolic system generator performs FSM-specific transformations. Finally, the **Final Optimization Module** (§3.4) integrates both paths and incorporates a verification system to ensure the functional correctness of the optimized design.

### 3.1 LLM Dispatcher

The **LLM Dispatcher** (②) receives the input RTL code and the specified optimization goal (*e.g.,* low power) (①) before any optimization begins. It first summarizes the code and generates potential optimization suggestions. These suggestions are then passed to the Retrieval-Augmented Generation (RAG) system to identify the relevant optimization rules. Additionally, the Dispatcher evaluates the presence of a Finite State Machine (FSM) in the original code to determine whether control flow optimization is necessary.

### 3.2 Data Flow Optimization

Data Flow represents the process by which information is propagated, processed, and optimized within an RTL design. Effective data flow optimizations improve system efficiency by simplifying computations, reducing redundancy, and enhancing PPA metrics. Common techniques include sub-

expression elimination, constant folding, and resource sharing. The proposed Data Flow Optimization Module addresses three key challenges: (1) generalizing diverse optimization patterns into accurate, reusable rules; (2) aligning LLM-generated optimizations with functional and logical requirements; and (3) resolving conflicts between optimization goals inherent in distinct design patterns.

**Optimization Rule Search Engine**   Optimization knowledge is often scattered across books, lecture notes, codebases, and design manuals, with no generalized repository to serve as a unified knowledge base. Furthermore, optimization patterns frequently *conflict* due to divergent goals, for example, power reduction versus performance improvement. To tackle these challenges, we developed a retrieval-augmented generation (RAG) system equipped with an optimization goal-based rule extraction module.

**Optimization Library.**   The RAG system aggregates raw RTL optimization data from sources such as lecture notes, manuals, and example designs [38, 35, 32, 30] into a comprehensive knowledge base (③). LLMs then summarize and structure these data into an optimization library (④). Each rule is abstracted to include its description, applicable optimization goals (*e.g.,* area, power, or timing), and its category (*e.g.,* data flow, FSM, MUX, memory, or clock gating). A similarity engine identifies overlaps with existing entries, prompting merges or exclusive labels to ensure the scalability of the rule library. To align optimization goals with generated outputs, the rules specify detailed instructions for constructing AST templates, enabling precise application of optimization patterns. Rules with clearly defined requirements include template-writing guidelines, while more abstract rules are stored as descriptive text and used directly as optimization prompts. See Appendix C for an example RAG.

**Enforcing Rule Alignment and Resolving Conflicts**   To improve the structure and alignment of optimization rules, the LLM Dispatcher (①) provides both a summary of the input RTL code and suggestions for potential optimizations. These inputs are passed to the search engine along with user-specified optimization goals, performing a similarity search to identify the most relevant rules from the RAG system.

Given the potential for conflicts between optimization goals, it is critical to prevent the inclusion of conflicting rules while ensuring no critical optimizations are overlooked. To achieve this balance, we employ the elbow method to analyze the similarity scores between the query and the candidate rules. This approach identifies a natural cutoff point where adding more rules no longer yields significant benefits. Let the similarity scores between the query and candidate rules be ordered as: $s_1 \geq s_2 \geq \cdots \geq s_M$, where $s_i$ denotes the similarity score for $i$-th rule, ordered from highest to lowest, and $M$ is the total number of candidates.

The optimal cutoff index $i^*$ is determined by maximizing the difference between consecutive similarity scores: $i^* = \underset{1 \leq i < M}{\arg\max}, (s_i - s_{i+1})$.

Rules with similarity scores above the threshold $\tau_{\text{elbow}}$ are selected for application. The similarity between the query embedding ($\mathbf{e}_{\text{query}}$) and the rule embedding ($\mathbf{e}_{\text{rule}}$) is computed as shown below:

$$\text{sim}(\mathbf{e}_{\text{query}}, \mathbf{e}_{\text{rule}}) = \frac{\mathbf{e}_{\text{query}} \cdot \mathbf{e}_{\text{rule}}}{|\mathbf{e}_{\text{query}}||\mathbf{e}_{\text{rule}}|} \geq \tau_{\text{elbow}}. \tag{1}$$

This method ensures that only the most relevant rules are selected, striking an optimal balance between comprehensiveness and precision. The output of the search engine contains two components: rules with detailed template-writing instructions (⑦) and abstract rules described only by their optimization properties (⑥).

**AST Template Building**   To ensure that LLM-generated RTL code aligns with functional and logical optimization goals, we enforce rules using AST-based symbolic systems, which have been proven to be effective in hardware debugging [37]. Compared to LLM-generated symbolic systems, AST-based templates offer several advantages: (1) parsing Verilog into an AST ensures accurate and structured design representations; (2) limiting each template to a single optimization goal maintains conciseness, facilitating correct generation and application by LLMs; and (3) the modular approach allows selection of templates to balance conflicting optimization patterns, enhancing flexibility.

For rules that include template-writing instructions, we prompt the LLM to generate an AST-based template that serves as a general optimization framework (⑧). Let $\mathcal{A}$ denote the set of all AST

nodes in the Verilog design, and let the matching condition: $\Phi : \mathcal{A} \to \{\text{true}, \text{false}\}$, determine whether a node qualifies for optimization. The process begins by identifying the **Target Node Type**, such as `Always`, `Instance`, `Assign` or `Module Instantiation`. For each node of the specified type, we apply $\Phi$ to decide whether it requires rewriting. Once the target nodes are identified, the **Transformation Rule** is applied as follows: $\tau : \{ a \in \mathcal{A} \mid \Phi(a) = \text{true} \} \longrightarrow \mathcal{A}$, where $\tau$ replaces the matched node with an optimized AST subtree (*e.g.,* merging nested `if-else` statements, folding constants, or simplifying expressions; code example see Appendix B). To ensure functional correctness, the transformed design undergoes an equivalence check using LLM-generated testbenches. If the template passes verification, it is stored in the RAG system as reusable content.

As shown in Figure 2, the RTL design is initially interpreted as an AST representation. The RAG system provides the LLM with multiple template options. Due to the varying optimization goals and scenarios, the system avoids relying on a fixed sequence of templates. Instead, the LLM determines which templates to apply and in what order, tailoring the optimization process to the design's specific requirements. To further prevent conflicts between templates or failures in the symbolic system, we introduce a **feedback loop**. This loop allows the LLM to re-select templates and adjust its strategy based on prior failures, ensuring robustness and adaptability.

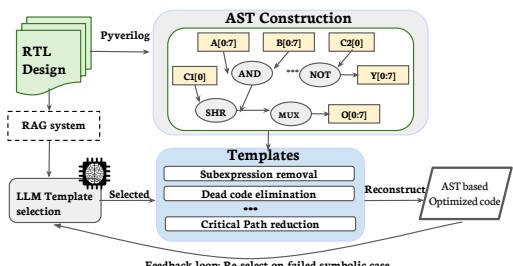

Figure 2: `SymRTLO` AST template optimization workflow.

## 3.3 Control Flow Optimization

Control Flow, unlike Data Flow's focus on how information is processed and propagated, defines the execution paths and sequencing of operations in RTL designs through finite-state machines (FSMs) that capture states, transitions, and outputs. These FSMs are tightly coupled with design constraints (*i.e.,* partial specifications, clock gating, and reset logic), making generic symbolic systems fragile or incomplete. Addressing these challenges requires deeper semantic analysis beyond simple pattern matching or generic AST templates. To enhance alignment between optimized code and the FSM minimization algorithm, we propose a Control Flow Optimization module utilizing an LLM-based symbolic system. An FSM can be formally represented as: $M = (Q, \Sigma, \delta, q_0, F)$, where $Q$ is the finite set of states, $\Sigma$ is the input alphabet, $\delta : Q \times \Sigma \to Q$ is the transition function, $q_0 \in Q$ is the initial state, and $F \subseteq Q$ is the set of accepting states.

For a partially specified FSM $M_p$, the transition function is extended to handle non-deterministic transitions: $\delta_p : Q \times \Sigma \to 2^Q$, where $2^Q$ represents the power set of $Q$.

Classical minimization algorithms (*e.g.,* Hopcroft's [18] or Moore's [28]) are effective for completely specified FSMs but are limited by real-world complexities. Practical RTL designs often integrate control logic with data path constraints, and undefined states and transitions make FSM minimization an NP-complete problem with a general complexity of $O(2^{|Q|})$. A single pre-built AST script cannot efficiently handle all such incomplete specifications. Let $\phi : Q \times D \to B$ represent the data path constraints, where $D$ is the data path state space and $B$ is the boolean domain. Pure FSM-focused AST-based optimization scripts can overlook these data path side effects, failing to capture deeper control semantics.

Inspired by [19], we propose leveraging LLMs to transform each circuit into a symbolic representation focused solely on FSM components, *i.e.,* isolating states, transitions, and relevant outputs, as illustrated in Figure 3. Instead of relying on a one-size-fits-all script, we prompt the LLM to dynamically generate a **specialized minimization script** tailored to the specific FSM structure and constraints. We show an example workflow in Appendix A.

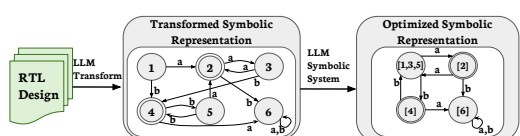

Figure 3: `SymRTLO` FSM optimization workflow.

### 3.4 Verification and Final Optimization

To address the challenges of manual verification and unreliable automated methods, we introduce an automated verification module that integrates functionality testing and formal equivalence checking. After AST-based optimization and the LLM-assisted symbolic system generate initial results, the LLM combines extracted rules, template-optimized code, and symbolic outputs to produce the final optimized RTL design ((11)). Verification is essential to ensure correctness, as LLM-generated rewrites may introduce unintended behavioral deviations. We employ a two-step verification pipeline: (1) the LLM generates test benches to validate basic functional correctness, acting as a rapid filter to reject invalid rewrites early; and (2) for designs that pass initial tests, we perform formal equivalence checking to formally confirm functional equivalence to the original design. For combinational logic and straightforward synchronous sequential circuits, we apply standard Boolean Satisfiability Problem (SAT)-based equivalence checking. While more complex designs like asynchronous resets, CDC paths, or retimed logic, we employ advanced sequential equivalence checking with robust state matching and transformation-tolerant verification techniques.

## 4 Experiments

### 4.1 Experimental Setup

**Baseline**   We compare `SymRTLO` with several state-of-the-art LLM and open source RTL optimization frameworks. The LLM baselines include GPT-O1, GPT-4o, GPT-4 [29], GPT-3.5[47], and GPT-4o-mini. Additionally, we include two specialized open-source LLM-based tools: Verigen [36] and RTL-Coder-Deepseek [25]. For a comprehensive evaluation, we analyze `SymRTLO` performance using circuits from the RTLRewriter Benchmark. Although the RTLRewriter environment is reproducible using Yosys [44] for wires and cells analysis, the exact test case they use is not provided. Moreover, comparing PPA results is even more challenging due to its reliance on Yosys + ABC [2]with unknown libraries. To demonstrate `SymRTLO`'s capabilities, we subject it to a broader evaluation scope, selecting examples that are diverse in size and functionality, while including cases reported in RTLRewriter's benchmark for direct comparison.

**Implementations**   The `SymRTLO` framework takes GPT-4o as its primary LLM for optimization strategy selection, symbolic system generation, and iterative HDL synthesis, leveraging its robust inference, low inference cost, and coding capabilities. While the framework is model-agnostic by design, alternatives like GPT-3.5 and GPT-O1 were excluded after preliminary tests showed poor results—GPT-3.5 lacked sufficient coding capability, and GPT-O1 incurred high latency and API costs, reducing overall efficiency. Pyverilog [34] is used for AST extraction and code reconstruction.

To efficiently retrieve relevant transformation templates and knowledge, we integrate OpenAI's text embedding-3-small, which excels in embedding-based retrieval tasks. For hardware compilation and validation, we use a combination of open-source and commercial tools. Yosys measures wires and cells, while Synopsys DC Compiler 2019 [33], paired with the Synopsys Standard Cell (SSC) library, performs PPA analysis. GPT-4o generates test benches for functional coverage. Yosys + ABC serves as the logical equivalence checker, while Synopsys Formality for sequential equivalence checking. For a fair comparison with standard compiler workflows, we apply typical Synopsys DC Compiler optimizations, using medium mapping effort and incremental mapping to reflect common practices.

**Evaluation Metrics**   First, to evaluate generation quality and functional correctness, we use the pass@k metric commonly employed in code generation tasks. This metric captures the probability that at least one valid solution exists within the top $k$ generations: pass@k $= \frac{1}{N} \sum_{i=0}^{N} \left(1 - \frac{C_{n_i-c_i}^{k}}{C_{n_i}^{k}}\right)$, where $N$ is the number of problems, $n_i$ and $c_i$ represent the total and correct samples for the $i$-th problem, respectively. Second, to test the performance of the synthesis results, we use the best results of the 10 valid generations of each model and our method. This sampling maintains token budget fairness, as SymRTLO with GPT-4o averages 7,728 tokens across all circuits, while the base GPT-4o model uses approximately 775 tokens per generation—a 10× difference.

For smaller benchmarks, we evaluate optimization results using Wires and Cells metrics, which reflect low-level physical characteristics of circuits. These metrics provide granular insights into routing complexity (wires) and logical component count (cells), offering a precise evaluation for isolated modules or blocks. For larger designs, we focus on PPA metrics to capture high-level efficiency and real-world applicability. These metrics offer a holistic view of resource usage and performance for complex designs, where low-level metrics like Wires and Cells become impractical.

Table 5: Comparison of Wire and Cell Counts. Yellow highlighting denotes state-of-the-art results. GeoMean is the geometric mean of the resource usage (wires or cells). Ratios are calculated by dividing the GeoMean by the baseline's resource usage. †: Reported Results from [46].

| Benchmark | Yosys Wires | Cells | GPT-4-Turbo Wires | Cells | GPT-4o Wires | Cells | GPT-3.5-Turbo Wires | Cells | GPT-4o-mini Wires | Cells | RTLCoder-DS Wires | Cells | RTLrewriter Wires | Cells | SymRTLO Wires | Cells |
|---|---|---|---|---|---|---|---|---|---|---|---|---|---|---|---|---|
| adder_subexpression | 8 | 3 | 7 | 3 | 7 | 3 | 7 | 3 | 7 | 3 | 8 | 3 | 7 | 3 | 7 | 3 |
| adder_architecture | 86 | 56 | 30 | 40 | 30 | 40 | 96 | 63 | 86 | 56 | 86 | 56 | - | - | 14 | 16 |
| multiplier_subexpr | 26 | 71 | 18 | 15 | 259 | 255 | 26 | 71 | 26 | 71 | 26 | 71 | - | - | 18 | 15 |
| constant_folding_raw | 12 | 6 | 10 | 5 | 10 | 5 | 10 | 5 | 10 | 5 | 12 | 6 | - | - | 8 | 5 |
| subexpression_elim | 17 | 12 | 19 | 12 | 19 | 12 | 19 | 12 | 17 | 10 | 17 | 12 | - | - | 14 | 8 |
| alu_subexpression | 30 | 24 | 30 | 24 | 28 | 22 | 30 | 24 | 27 | 22 | 30 | 24 | 21 | 18 | 21 | 18 |
| adder_resource | 13 | 3 | 6 | 3 | 9 | 4 | 6 | 3 | 7 | 3 | 13 | 3 | - | - | 6 | 3 |
| multiplier_bitwidth | 9 | 3 | 8 | 3 | 8 | 3 | 9 | 3 | 9 | 3 | 9 | 3 | 8 | 3 | 8 | 3 |
| multiplier_architect | 4 | 2 | 14 | 36 | 4 | 2 | 16 | 20 | 18 | 36 | 4 | 2 | - | - | 4 | 2 |
| adder_bit_width | 4 | 1 | 3 | 1 | 3 | 1 | 3 | 1 | 3 | 1 | 4 | 1 | 3 | 1 | 3 | 1 |
| loop_tiling_raw | 5 | 16 | 4 | 16 | 4 | 16 | 4 | 16 | 484 | 496 | 5 | 16 | - | - | 3 | 16 |
| **GeoMean** | 15.49 | 8.96 | 13.45 | 9.81 | 16.35 | 9.97 | 16.40 | 11.49 | 16.31 | 11.49 | 15.49 | 8.96 | - | - | 9.75 | 5.95 |
| **Ratio** | 1.00 | 1.00 | 0.87 | 1.10 | 1.06 | 1.11 | 1.06 | 1.28 | 1.11 | 1.31 | 1.15 | 0.91 | 0.69†1 | 0.77†1 | 0.63 | 0.67 |

Table 6: FSM Designs PPA Comparison. Yellow highlights indicate state-of-the-art results. A ⇓ marks improvement, while a ⇑ denotes a decline compare with the original design. Two comparison scenarios are shown: without compiler optimization (upper improvement) and with compiler optimization (lower improvement). A - indicates that no code is available for analysis.

| Model/Method | example1_state Power (mW) | Time (ns) | Area ($\mu m^2$) | example2_state Power (mW) | Time (ns) | Area ($\mu m^2$) | example3_state Power (mW) | Time (ns) | Area ($\mu m^2$) | example4_state Power (mW) | Time (ns) | Area ($\mu m^2$) | example5_state Power (mW) | Time (ns) | Area ($\mu m^2$) |
|---|---|---|---|---|---|---|---|---|---|---|---|---|---|---|---|
| Original | 0.042 | 1.21 | 833.0 | 0.056 | 2.25 | 549.4 | 0.052 | 1.35 | 589.6 | 0.055 | 2.18 | 597.1 | 0.055 | 2.18 | 597.1 |
| GPT-3.5 | 0.043 | 1.27 | 870.6 | 0.056 | 2.25 | 549.4 | 0.052 | 1.35 | 589.6 | 0.059 | 2.17 | 972.3 | 0.055 | 2.18 | 597.1 |
| GPT4o-mini | 0.055 | 1.08 | 1021.1 | 0.062 | 2.23 | 579.2 | 0.063 | 1.08 | 714.9 | 0.055 | 2.18 | 597.1 | 0.053 | 2.18 | 634.7 |
| GPT-4-Turbo | 0.053 | 2.97 | 993.5 | 0.067 | 2.28 | 737.6 | 0.065 | 1.22 | 810.3 | 0.055 | 2.18 | 273.5 | 0.029 | 2.25 | 366.8 |
| GPT-4o | 0.053 | 2.97 | 1002.5 | 0.056 | 2.25 | 549.4 | 0.052 | 1.35 | 589.6 | 0.055 | 2.18 | 273.5 | 0.055 | 2.18 | 597.1 |
| GPT-O1 | 0.044 | 1.17 | 910.7 | 0.056 | 2.25 | 549.4 | 0.052 | 1.35 | 589.6 | 0.055 | 2.18 | 597.1 | 0.055 | 2.18 | 597.1 |
| RTLCoder-DS | 0.042 | 1.21 | 833.0 | 0.056 | 2.25 | 549.4 | 0.052 | 1.35 | 589.6 | 0.055 | 2.18 | 597.1 | 0.063 | 2.34 | 649.78 |
| RTLrewriter | 0.025 | 3.23 | 424.0 | - | - | - | 0.041 | 1.36 | 549.4 | - | - | - | - | - | - |
| SymRTLO | 0.029 | 1.21 | 403.9 | 0.024 | 1.17 | 271.0 | 0.023 | 1.15 | 268.5 | 0.024 | 2.17 | 273.5 | 0.026 | 2.18 | 270.9 |
| **Improvement(%)** | ⇓30.95 | 0.00 | ⇓51.51 | ⇓57.14 | ⇓48.00 | ⇓50.67 | ⇓55.77 | ⇓14.81 | ⇓54.46 | ⇓56.36 | ⇓0.46 | ⇓54.20 | ⇓52.73 | 0.00 | ⇓54.63 |
| Original + Compiler Opt. | 0.041 | 2.85 | 564.51 | 0.035 | 2.24 | 316.11 | 0.038 | 1.23 | 358.77 | 0.044 | 2.19 | 451.59 | 0.045 | 2.19 | 451.59 |
| SymRTLO + Compiler Opt. | 0.021 | 2.64 | 240.85 | 0.018 | 0.6 | 175.61 | 0.018 | 2.42 | 180.63 | 0.020 | 2.17 | 185.65 | 0.019 | 2.27 | 188.169 |
| **Improvement(%)** | ⇓48.78 | ⇓7.37 | ⇓57.33 | ⇓48.57 | ⇓73.21 | ⇓44.45 | ⇓52.63 | ⇑96.74 | ⇓49.65 | ⇓54.55 | ⇓0.91 | ⇓58.89 | ⇓57.78 | ⇑3.65 | ⇓58.33 |

## 4.2 Functional Correctness Analysis

To demonstrate that our method reduces synthesis time and improves functional correctness, Table 4 presents the evaluation results. SymRTLO achieves near-perfect first-attempt pass rates, ensuring valid, optimized RTL code with maintained functional equivalence. This significantly outperforms state-of-the-art language models, particularly given the complexity of RTL optimization tasks and the necessity of maintaining functional equivalence. SymRTLO reduces synthesis iterations, minimizing redundant computations throughout the optimization process.

Table 4: Pass Rate Results.

| Method | Pass@1 | Pass@5 | Pass@10 |
|---|---|---|---|
| Ours | **97.5** | **100.0** | **100.0** |
| GPT-4o | 45.9 | 60.0 | 72.7 |
| GPT-4-Turbo | 42.9 | 62.7 | 81.8 |
| GPT-4o-mini | 2.5 | 10.9 | 12.7 |
| GPT-3.5-Turbo | 28.6 | 42.7 | 54.5 |
| RTL-Coder DeepSeek | 8.8 | 18.2 | 27.3 |
| Verigen-2B | 0.0 | 0.0 | 0.0 |
| Verigen-16B | 0.0 | 0.0 | 0.0 |

## 4.3 Circuit Optimization Performance

To demonstrate SymRTLO's effectiveness in resolving cross-rule conflicts and achieving optimization objectives, we conduct an experiment presented in Figure 4. Given the limited benchmarks available for LLM-driven RTL design, we analyze our framework on RTLRewriter's benchmark, which primarily emphasizes area optimization, leaving minimal room for improvements in power and delay. To align with this limitation, we put area opti-

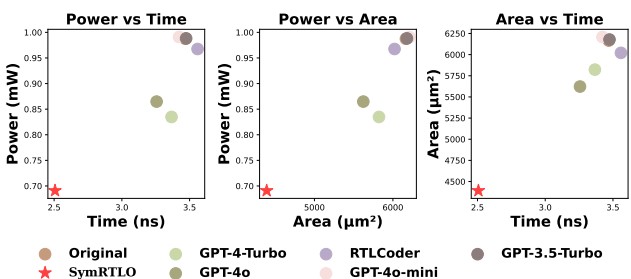

Figure 4: The PPA overall improvement of benchmark cases.

Table 7: Algorithm Optimizations PPA Comparison. Yellow highlights indicate state-of-the-art results. A ⇓ marks improvement, while a ⇑ denotes a decline compare with the original design. Two comparison scenarios are shown: without compiler optimization (upper improvement) and with compiler optimization (lower improvement).

| Model/Method | sppm_redundancy | | | subexpression_elim | | | adder_architecture | | | vending | | | fft | | |
|---|---|---|---|---|---|---|---|---|---|---|---|---|---|---|---|
| | Power (mW) | Time (ns) | Area ($\mu m^2$) | Power (mW) | Time (ns) | Area ($\mu m^2$) | Power (mW) | Time (ns) | Area ($\mu m^2$) | Power (mW) | Time (ns) | Area ($\mu m^2$) | Power (mW) | Time (ns) | Area ($\mu m^2$) |
| Original | 2.86 | 7.41 | 40102.6 | 5.27 | 11.09 | 10989.1 | 0.418 | 2.78 | 1023.5 | 7.61 | 227.86 | 176982.98 | 58.23 | 8.26 | 2255264.75 |
| GPT-3.5-Turbo | 2.86 | 7.41 | 40102.6 | 5.27 | 11.09 | 10989.1 | 0.418 | 2.78 | 1023.5 | 7.61 | 227.86 | 176982.98 | 58.23 | 8.26 | 2255264.75 |
| GPT4o-mini | 2.86 | 7.41 | 40102.6 | 5.27 | 11.09 | 10989.1 | 0.418 | 2.78 | 1023.5 | 7.61 | 227.86 | 176982.98 | 58.23 | 8.26 | 2255264.75 |
| GPT-4-Turbo | 2.86 | 7.41 | 40102.6 | 3.93 | 11.09 | 7783.03 | 0.392 | 2.74 | 1023.5 | 7.50 | 227.86 | 176982.98 | 58.23 | 8.26 | 2255264.75 |
| GPT-4o | 2.86 | 7.41 | 40102.6 | 5.27 | 11.09 | 8984.61 | 0.392 | 2.74 | 1023.5 | 7.61 | 227.86 | 176982.98 | 58.23 | 8.26 | 2255264.75 |
| GPT-O1 | 1.87 | 7.41 | 29919.8 | 4.63 | 11.09 | 8957.23 | 0.418 | 2.78 | 1023.52 | 7.61 | 227.86 | 176982.98 | 58.23 | 8.26 | 2255264.75 |
| RTLCoder-DS | 2.86 | 7.41 | 40102.6 | 5.27 | 11.09 | 10989.1 | 0.418 | 2.78 | 1023.5 | 7.61 | 227.86 | 176982.98 | 58.23 | 8.26 | 2255264.75 |
| SymRTLO | **1.77** | **7.29** | **29606.18** | **3.02** | **2.87** | **7358.8** | **0.328** | **1.97** | **762.6** | **6.97** | 227.86 | **164831.1** | **31.71** | **8.09** | **1726125.71** |
| **Improvement(%)** | ⇓38.46 | ⇓1.62 | ⇓26.17 | ⇓42.68 | ⇓74.12 | ⇓33.04 | ⇓21.53 | ⇓29.14 | ⇓25.49 | ⇓8.41 | ⇓0 | ⇓6.87 | ⇓45.54 | ⇓2.06 | ⇓23.46 |
| Original + Compiler Opt. | 1.46 | 7.95 | 22908.69 | 4.61 | 11.78 | 9484.15 | 0.17 | 2.29 | 541.92 | 11.46 | 7.90 | 240079.29 | 51.12 | 7.90 | 1857805.49 |
| SymRTLO + Compiler Opt. | 1.46 | 7.95 | 22908.69 | **3.53** | 11.78 | **6791.88** | 0.17 | 2.48 | **531.88** | **8.175** | 7.90 | **151593.86** | **26.32** | 8.98 | **1471378.46** |
| **Improvement(%)** | 0 | 0 | 0 | ⇓23.4 | 0 | ⇓28.39 | 0 | ⇑8.30 | ⇓1.86 | ⇓28.66 | 0 | ⇓36.86 | ⇓48.51 | ⇑13.67 | ⇓20.8 |

mization as our primary goal. Despite these constraints, SymRTLO achieves substantial improvements, averaging 40.96% in power, 17.02% in delay, and 38.05% in area, while maintaining a balanced optimization across all three metrics, highlighting its versatility and robustness.

We present a comprehensive analysis of SymRTLO capabilities using 11 short-benchmark examples from RTLRewriter focusing on Wires and Cells optimization, along with 10 complex FSM and algorithm examples from both short and long benchmarks. This representative selection demonstrates SymRTLO's scalability and effectiveness across diverse functional domains but also aligns with circuits reported in RTLRewriter's paper, enabling direct comparison with state-of-the-art results.

Smaller benchmarks requiring only 1-2 optimization patterns provide ideal test cases for LLM output *alignment*. Table 5 shows that SymRTLO consistently outperforms baseline implementations across various test cases. With wire and cell ratios of 0.63 and 0.67 respectively, it surpasses the state-of-the-art values of 0.69 and 0.77. While models like GPT-4 excel in certain cases, they lack consistency across diverse optimization tasks.

In FSM PPA experiments, SymRTLO significantly outperforms existing approaches, particularly in relation to RTLRewriter, the state-of-the-art solution, achieving an improvement of up to 50.59%, 12.65%, 53.09% in power, time, and area, respectively. As shown in Table 6, it effectively aligns the FSM state reduction algorithm with optimized code, minimizing all FSM states and achieving the best overall PPA results. This demonstrates that the LLM-generated symbolic system is both stable and aligned with intended optimization goals.

To evaluate the effectiveness of generalized rules and AST templates in balancing conflicting rules, we conduct algorithm case PPA experiments involving complex Data Path and Control Path scenarios. As shown in Table 7, SymRTLO applies AST templates, optimized rules, and minimized FSM states, achieving 30.34%, 21.37%, and 20.01% improvements in PPA over GPT-4o, our base model, on average. Note that RTLRewriter's results are unavailable for comparison for these cases.

We test SymRTLO with Synopsys DC optimization workflows with medium mapping effort, incremental mapping for both FSM and Algorithm cases, as shown in Table 6 and Table 7, demonstrating further balanced optimization alongside compiler optimization processes, achieving overall improvements of 36.2% in power and 35.66% in area, with only an 8.3% increase in time as a trade-off. Even under more stringent compiler optimization settings of flattened mode with high mapping effort, SymRTLO still delivers overall improvements of 27.7% in area, 35.8% in power, and 0.5% in delay.

## 4.4 Ablation Studies

To assess the effectiveness of individual components in SymRTLO, we conduct ablation studies by systematically removing one component at a time from the complete SymRTLO framework. In the first setting, disabling the AST-based template generation ("Remove Template-Based Opt") forced the model to rely solely on abstract rule descriptions without structural guidance. In the second, removing the FSM symbolic optimization module ("Remove Symbolic Reasoning") limited the framework to dataflow-only optimization.

In the third, turning off the goal-based rule filtering ("Remove Goal-Based Search") allowed all retrieved rules to be applied regardless of potential conflicts. For each configuration, we measured average PPA improvements across all benchmarks relative to the original designs. Figure 5 summarizes these results, showing that all three components contribute significantly to SymRTLO's overall performance. Removing any one of them results in substantial losses in optimization effectiveness, further emphasizing the necessity of their integration. By contrast, GPT-4o alone achieves minimal improvements, underscoring the advantages of SymRTLO's tailored framework.

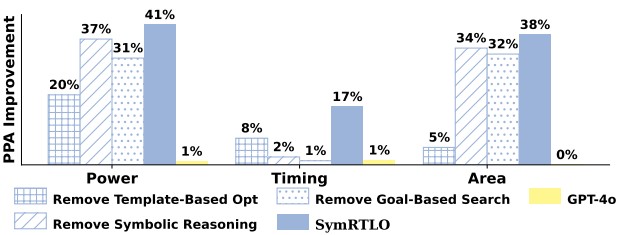

Figure 5: Ablation study showing average PPA improvements when individual SymRTLO components are removed. Each bar represents the framework's performance with one component disabled, demonstrating the necessity of all three components for optimal results.

## 5 Discussion

It is important to clarify that SymRTLO operates as a *pre-synthesis optimizer* that complements rather than replaces traditional synthesis tools. RTL logic synthesis (e.g., Synopsys Design Compiler) takes RTL hardware descriptions and converts them to optimized gate-level netlists through optimizations including Boolean logic simplification, constant propagation, technology mapping, gate sizing, and retiming. While synthesis tools can perform local optimizations and limited resource sharing, they generally preserve the fundamental RTL architecture and control flow structure—operating within the constraints of the original RTL specification.

In contrast, SymRTLO performs *source-level code optimization* that modifies the RTL before synthesis, enabling algorithmic transformations, architectural restructuring, dataflow modifications, FSM redesign, and pipeline adjustments that fundamentally alter the hardware implementation. By optimizing at the RTL source level, SymRTLO can explore a broader design space and make architectural decisions that are beyond the scope of gate-level synthesis tools, typically yielding larger PPA improvements than synthesis optimizations alone.

## 6 Limitations

While SymRTLO demonstrates significant improvements in RTL optimization, the framework's performance is fundamentally dependent on the underlying capabilities of the base LLM, which may limit its effectiveness on novel or highly specialized circuit patterns not well-represented in its training data. The optimization rule library, though comprehensive, requires ongoing maintenance and expansion to keep pace with evolving hardware design practices and emerging optimization techniques. Since final PPA evaluation relies on synthesis tools like Synopsys DC Compiler and Yosys, results may vary across different tool versions or technology libraries. Additionally, while the symbolic reasoning module effectively handles FSM optimization, the approach may face scalability challenges when applied to extremely large state machines or circuits with highly complex control flows that exceed current computational bounds for formal analysis.

## 7 Conclusion

We present SymRTLO, a neuron-symbolic framework that integrates LLM-based code rewriting and symbolic reasoning to optimize both data flow and control flow in RTL designs. SymRTLO generalizes optimization rules, aligns generated code with intended transformations, resolves conflicting optimization goals, and ensures reliable automated verification. By combining retrieval-augmented guidance with symbolic systems, SymRTLO automates complex structural rewrites while maintaining functional correctness. Extensive evaluations on industrial-scale designs demonstrate significant PPA gains over state-of-the-art solutions.

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

# A  FSM Symbolic System

The following section demonstrates an example FSM in verilog. First the verilog is transformed to
Symbolic Representation, then the Symbolic system applied minimization algorithm to optimize the
FSM.

```verilog
module example(
    input wire clk,
    input wire reset,
    input wire [1:0] input_signal,
    output reg output_signal);
    parameter S0 = 3'b000,S1 = 3'b001,S2 = 3'b010,S3 = 3'b011,S4 = 3'b100,S5 =
    ↪    3'b101;
    reg [2:0] current_state, next_state;
    always @(current_state) begin
        output_signal = 0;
        case (current_state)
            S0: output_signal = 1;
            S2: output_signal = 1;
            S4: output_signal = 1;
            default: output_signal = 0;
        endcase
    end
    always @(posedge clk or posedge reset) begin
        if (reset) begin
            current_state <= S0; // Reset to state S0
        end else begin
            current_state <= next_state;
        end
    end
    always @(*) begin
        next_state = current_state;
        case (current_state)
            S0: case (input_signal)
                2'b00: next_state = S0;
                2'b01: next_state = S1;
                2'b10: next_state = S2;
                2'b11: next_state = S3;
            endcase
            S1: case (input_signal)
                2'b00: next_state = S0;
                2'b01: next_state = S3;
                2'b11: next_state = S5;
            endcase
            S2: case (input_signal)
                2'b00: next_state = S1;
                2'b01: next_state = S3;
                2'b10: next_state = S2;
                2'b11: next_state = S4;
            endcase
            S3: case (input_signal)
                2'b00: next_state = S1;
                2'b01: next_state = S0;
                2'b10: next_state = S4;
                2'b11: next_state = S5;
            endcase
            S4: case (input_signal)
                2'b00: next_state = S0;
                2'b01: next_state = S1;
                2'b10: next_state = S2;
                2'b11: next_state = S5;
            endcase
            S5: case (input_signal)
                2'b00: next_state = S1;
                2'b01: next_state = S4;
                2'b10: next_state = S0;
            endcase
        endcase
    end
endmodule
```

*Listing 1:* Example Test Case: example1_state.

```
1 'states': ['S0', 'S1', 'S2', 'S3', 'S4', 'S5'],
2 'transitions': {
3     'S0': {
4         'input_signal=00': {'next_state': 'S0'},
5         'input_signal=01': {'next_state': 'S1'},
6         'input_signal=10': {'next_state': 'S2'},
7         'input_signal=11': {'next_state': 'S3'}
8     },
9     'S1': {
10        'input_signal=00': {'next_state': 'S0'},
11        'input_signal=01': {'next_state': 'S3'},
12        'input_signal=11': {'next_state': 'S5'}
13    },
14    'S2': {
15        'input_signal=00': {'next_state': 'S1'},
16        'input_signal=01': {'next_state': 'S3'},
17        'input_signal=10': {'next_state': 'S2'},
18        'input_signal=11': {'next_state': 'S4'}
19    },
20    'S3': {
21        'input_signal=00': {'next_state': 'S1'},
22        'input_signal=01': {'next_state': 'S0'},
23        'input_signal=10': {'next_state': 'S4'},
24        'input_signal=11': {'next_state': 'S5'}
25    },
26    'S4': {
27        'input_signal=00': {'next_state': 'S0'},
28        'input_signal=01': {'next_state': 'S1'},
29        'input_signal=10': {'next_state': 'S2'},
30        'input_signal=11': {'next_state': 'S5'}
31    },
32    'S5': {
33        'input_signal=00': {'next_state': 'S1'},
34        'input_signal=01': {'next_state': 'S4'},
35        'input_signal=10': {'next_state': 'S0'}
36    }
37 },
38 'outputs': {'S0': {'output_signal': 1},
39             'S1': {'output_signal': 0},
40             'S2': {'output_signal': 1},
41             'S3': {'output_signal': 0},
42             'S4': {'output_signal': 1},
43             'S5': {'output_signal': 0} }
```

*Listing 2:* LLM-generated symbolic representation for example1_state.

```
1 State: S2, Output: 1
2   input_signal=00 -> S1
3   input_signal=01 -> S3_S5
4   input_signal=10 -> S2
5   input_signal=11 -> S0_S4
6 State: S0_S4, Output: 1
7   input_signal=00 -> S0_S4
8   input_signal=01 -> S1
9   input_signal=10 -> S2
10  input_signal=11 -> S3_S5
11 State: S1, Output: 0
12  input_signal=00 -> S0_S4
13  input_signal=01 -> S3_S5
14  input_signal=10 -> S1
15  input_signal=11 -> S3_S5
16 State: S3_S5, Output: 0
17  input_signal=00 -> S1
18  input_signal=01 -> S0_S4
19  input_signal=10 -> S0_S4
20  input_signal=11 -> S3_S5
```

*Listing 3:* Reduced states of example1_state.

# B AST Template

The following section demonstrate how applying AST templates transforms the code and apply optimization patterns.

```
1 module example_raw
2 #(  parameter       BW = 8)
3 (
4     input [BW-1:0] a,
5     input [BW-1:0] b,
6     input [BW-1:0] c,
7     input [BW-1:0] d,
8     output [BW-1:0] s1
9 );
10     assign s2 = a * b;
11     assign s3 = a % b +d;
12     assign s4 = c + d + b * a;
13     assign s5 = a - b;
14     assign s6 = (b + 1) * a + d + c -b;
15     assign s1 = a + 23;
16 endmodule
```

*Listing 4:* Example Test Case: dead_code_elimination.

```
1 module example_raw #
2 (parameter BW = 8)
3 (
4   input [BW-1:0] a,
5   input [BW-1:0] b,
6   input [BW-1:0] c,
7   input [BW-1:0] d,
8   output [BW-1:0] s1
9 );
10   assign s1 = a + 23;
11 endmodule
12
```

*Listing 5:* Example Test Case: dead_code_elimination after applying the Dead Code Elimination AST template.

```
1 module example_raw
2 #(  parameter       BW = 8)
3 (
4     input [BW-1:0] a,
5     input [BW-1:0] b,
6     input [BW-1:0] c,
7     input [BW-1:0] d,
8     output [BW-1:0] s1,
9     output [BW-1:0] s2,
10    output [BW-1:0] s3,
11    output [BW-1:0] s4,
12    output [BW-1:0] s5,
13    output [BW-1:0] s6
14 );
15     assign s1 = a + b;
16     assign s2 = a * b;
17     assign s3 = a \% b +d;
18     assign s4 = c + d + b * a;
19     assign s5 = a - b;
20     assign s6 = (b + 1) * a + d + c -b;
21 endmodule
```

*Listing 6:* Example Test Case: subexpression_elimination.

```
1 module example
2 #(  parameter      BW = 8)
3 (   input [BW-1:0] a,
4     input [BW-1:0] b,
5     input [BW-1:0] c,
6     input [BW-1:0] d,
7     output [BW-1:0] s1,
8     output [BW-1:0] s2,
9     output [BW-1:0] s3,
10    output [BW-1:0] s4,
11    output [BW-1:0] s5,
12    output [BW-1:0] s6
13 );
14    assign s1 = a + b;
15    assign s2 = a * b;
16    assign s3 = a \% b + d;
17    assign s4 = c + d + s2;
18    assign s5 = a - b;
19    assign s6 = s4 + s5;
20 endmodule
```

*Listing 7:* Example Test Case: subexpression_elimination after applying the Common Sub-Expressions Elimination template. The Common Sub-Expressions are reused in the states after it.

```
1 module example_raw
2 #(  parameter      BW = 8)
3 (
4     input [BW-1:0] a,
5     input [BW-1:0] b,
6     output [BW-1:0] s1,
7     output [BW-1:0] s2
8 );
9     wire [BW-1:0] t1, t2;
10    assign s1 = a + b;
11    assign t1 = s1 + 0;
12    assign t2 = s1 * 1;
13    assign s2 = t1 + t2;
14 endmodule
15
```

*Listing 8:* Example Test Case: algebraic_simplification.

```
1
2 module example_raw #
3 (
4   parameter BW = 8
5 )
6 (
7   input [BW-1:0] a,
8   input [BW-1:0] b,
9   output [BW-1:0] s1,
10  output [BW-1:0] s2
11 );
12
13   assign s1 = a + b;
14   assign s2 = s1 + s1;
15
16 endmodule
```

*Listing 9:* Example Test Case: algebraic_simplification after applying the Temporary Variable Elimination, Dead Code Elimination, then Expression Simplification templates.

## C  RAG Example

---

**Sample Retrieval Augmented Optimization Rule**

"name": "IntermediateVariableExtraction",
"pattern": "Detect conditional assignments to a register based on a control signal",
"rewrite": "Extract common sub-expressions into intermediate variables to reduce redundant logic",
"category": "combinational/dataflow",
"objective_improvement": "area",
"template_guidance": "To implement this rule in a Python template subclassing BaseTemplate, use pyverilog AST manipulation to identify conditional assignments (vast.IfStatement) and extract the common sub-expressions into separate assignments. Look for vast.Identifier nodes that are assigned conditionally and create new vast.Assign nodes for the intermediate variables. Ensure that the new assignments are placed before the conditional logic to maintain correct data flow",
"function_name": "IntermediateVariableExtractionTemplate"

---

Figure 6: Sample Retrieval Augmented Optimization Rule 1: Intermediate Variable Extraction.

---

**Sample Retrieval Augmented Optimization Rule**

"name": "HierarchicalExpressionTreeRestructuring",
"pattern": "Detect flat conditional assignments with redundant arithmetic operations across different control paths",
"rewrite": "Transform into hierarchical tree structure with intermediate wire assignments for resource sharing",
"category": "combinational/dataflow",
"objective_improvement": "area, delay",
"template_guidance": "Parse conditional assignments, extract common subexpressions (e.g., 'y0_real * y0_real + y0_imag * y0_imag'), create intermediate wires, and build balanced adder trees. Replace flat expressions with hierarchical structure",
"function_name": "HierarchicalTreeRestructuringTemplate"

---

Figure 7: Sample Retrieval Augmented Optimization Rule 2: Hierarchical Expression Tree Restructuring.

---

**Sample Retrieval Augmented Optimization Rule**

"name": "AlgorithmicLoopToPipelineTransformation",
"pattern": "Detect sequential loops with independent operations where each iteration processes different data elements without dependencies",
"rewrite": "Transform sequential loops into pipelined architectures with staged data flow and parallel processing units",
"category": "control/dataflow",
"objective_improvement": "delay, power",
"template_guidance": "Identify loops like 'for(i=0; i<N; i++) output[i] = func(input[i])' with no loop-carried dependencies. Convert to pipeline stages where each stage processes one element: create N pipeline registers, instantiate processing units between stages, add valid/ready handshaking. Replace loop with continuous data flow through pipeline stages",
"function_name": "LoopToPipelineTemplate"

---

Figure 8: Sample Retrieval Augmented Optimization Rule 3: Algorithmic Loop-to-Pipeline Transformation.

> **Sample Retrieval Augmented Optimization Rule**
>
> "name": "Zero Multiplication Elimination",
> "pattern": "Detect multiplication by zero in expressions (*e.g.,* , 0 * c)",
> "rewrite": "Eliminate multiplication by zero, replacing the entire expression with zero",
> "category": "combinational/dataflow",
> "objective_improvement": "area",
> "template_guidance": "Identify vast.Times nodes with a zero operand. Replace the node with a vast.IntConst node representing zero.",
> "function_name": "ZeroMultiplicationTemplate"

Figure 9: Sample Retrieval Augmented Optimization Rule 4: Zero Multiplication Rule.

> **Hardware Optimization Rule**
>
> "name": "ReplaceRippleCarryWithCarryLookahead",
> "pattern": "Detects a ripple carry adder implementation using a series of full adders connected in sequence",
> "rewrite": "Transforms the ripple carry adder into a carry lookahead adder by using partial full adders and generating carry bits in parallel",
> "category": "combinational/dataflow",
> "objective_improvement": "area, delay",
> "template_guidance": null,
> "function_name": null

Figure 10: Sample Retrieval Augmented Optimization Rule 5: Replace Ripple Carry with Carry Lookahead, no template guidance is needed since it is an abstract rule.

> **Sample Retrieval Augmented Optimization Rule**
>
> "name": "AlgorithmAwareBitwidthOptimization",
> "pattern": "Identify oversized register widths in arithmetic pipelines with scaling factors",
> "rewrite": "Reduce bit-widths based on mathematical analysis of scaling requirements while preserving precision",
> "category": "datapath/precision",
> "objective_improvement": "area, power",
> "template_guidance": "Analyze scaling factors in arithmetic operations, calculate minimum required bit-widths based on dynamic range analysis, modify register declarations and shift operations. Example: reduce 40-bit to 28-bit when 13-bit scaling is excessive. Use pyverilog to identify Width nodes and replace with optimized bit-width declarations",
> "function_name": "BitwidthOptimizationTemplate"

Figure 11: Sample Retrieval Augmented Optimization Rule 6: Algorithm-Aware Bit-width Optimization.

