# OpenReview forum: "SymRTLO: Enhancing RTL Code Optimization with LLMs and Neuron-Inspired Symbolic Reasoning"
_NeurIPS.cc/2025/Conference — NeurIPS 2025 poster_

### Official Review · Reviewer_b1CD · 2025-06-27

**Clarity:** 2
**Significance:** 3
**Originality:** 2
**Rating:** 4
**Confidence:** 3

**Summary:**

This paper presents SymRTLO, a neuro-symbolic framework that combines large language models (LLMs) with symbolic reasoning for optimizing Register Transfer Level (RTL) code. The key contributions include: (1) a retrieval-augmented system for extracting and applying optimization rules, (2) an AST-based template approach to maintain syntactic correctness, and (3) a symbolic module for finite state machine (FSM) optimization. The authors demonstrate improvements in power, performance, and area (PPA) over state-of-the-art methods, including both traditional compiler-based tools and LLM-driven approaches. The framework aims to reduce reliance on iterative synthesis feedback while improving alignment between optimization goals and generated RTL code.

**Questions:**

- Can the authors clarify where LLMs provide unique advantages over traditional symbolic methods?
- The completeness of LLM-generated testbenches (e.g., branch/state coverage)?
- How false positives/negatives in equivalence checking are mitigated?
- The RAG examples (Appendix C) include simplistic rules. Are there more complex, LLM-specific optimizations that traditional tools cannot easily replicate?
- The paper acknowledges LLM reliability issues but relies on post-hoc verification. Are there safeguards during code generation (e.g., constrained decoding, symbolic grounding)?

**Ethical Concerns:**

["NO or VERY MINOR ethics concerns only"]

**Final Justification:**

The rebuttal has addressed most of my concerns. I now acknowledge that this paper is acceptable, conditioned on providing more unique capabilities over conventional synthesis tools.

**Limitations:**

yes

**Paper Formatting Concerns:**

There are minor issues. Figure 3 is floating over page margin. Spaces around tables are aggressively reduced.

**Quality:**

1

**Strengths And Weaknesses:**

Strengths:
- Integrating LLM into compilation is a promising direction to explore.
- Introduced various new technologies into an old problem, e.g. LLM, RAG, symbolic reasoning etc.

Weaknesses:
- Many optimizations presented (e.g., dead code elimination, subexpression removal) are standard in existing EDA tools. The paper does not sufficiently justify why LLMs are needed for tasks that deterministic algorithms already solve reliably.
- The RAG-based rule extraction (Appendix C) includes trivial optimizations (e.g., "zero multiplication elimination") that do not clearly benefit from LLM involvement. A stronger case should be made for where LLMs provide unique value beyond traditional methods.
- The paper overlooks the reliability risks of introducing generative models into compilation. The paper claims a "fast verification pipeline," but testbench generation is a long-standing complex problem that is not solvable in a paragraph of a NeurIPS paper. The current approach (LLM-generated test cases + equivalence checking) seems amateurish.

---

> ### Author Rebuttal · Authors · 2025-07-31
>
> We sincerely thank the reviewer for their thorough evaluation and constructive feedback. Your detailed technical questions have provided valuable opportunities to clarify our methodology and demonstrate additional experimental evidence. We believe our comprehensive responses address the raised concerns and showcase the meaningful impact SymRTLO can have on the RTL optimization problem. We hope this enhanced understanding of our work merits consideration for an improved evaluation score.
>
> ## Q1: Can the authors clarify where LLMs provide unique advantages over traditional symbolic methods?
>
> LLMs provide unique advantages in **source-level RTL code optimization**, which operates fundamentally differently from synthesis-level optimization. Traditional EDA tools optimize at the gate-level netlist after synthesis, while our approach optimizes the RTL source code itself, making it **tool-independent** and more **explainable and customizable** to engineers. Rather than replacing traditional methods, SymRTLO works as an **addition to traditional synthesis optimization**. As shown in Table 6-7 (lines 379-381), "Even under more stringent compiler optimization settings of flattened mode with high mapping effort, SymRTLO still delivers overall improvements of 27.7% in area, 35.8% in power, and 0.5% in delay."
>
> As explained in Section 2, compiler-based methods **"rely heavily on predefined heuristics, making them ill-suited for adapting to unconventional and emerging design patterns”**,  while SymRTLO combines RAG with LLM-driven symbolic reasoning for dynamic, goal-based optimization.
>
> ### Key LLM Advantages:
> - **Cross-Domain Pattern Recognition:** LLMs simultaneously understand algorithmic intent and hardware constraints.
>
> - **Context-Aware FSM Optimization:** Unlike fixed algorithms (Hopcroft, Moore), LLMs generate specialized minimization strategies for specific designs.
>
> - **High-Level Pattern Recognition:** LLMs identify algorithmic structures invisible to traditional pattern-matching tools.
>
> - **Dynamic Rule Orchestration:** LLMs adaptively select and sequence optimizations based on design characteristics and conflicting goals (power vs. area vs. timing), unlike predetermined compiler passes.
>
> - **Pattern Scalability:** LLMs generalize to handle emerging RTL design patterns by expanding the knowledge base.
>
> ## Q2: Testbench Coverage and Verification Methodology
>
> **We acknowledge and agree that testbench generation remains an important research area.** We want to clarify that in SymRTLO, **testbenches serve as rapid initial filters for invalid designs**, while formal verification tools (Synopsys Formality) provide actual correctness guarantees through FSM equivalence checking. LLM-generated tests balance computational efficiency and low latency with comprehensive coverage, as formal verification ensures final correctness; the testbenches are typically generated under 5 attempts.
>
> ### Tier 1: Testbench
>
> We have conducted experiments on coverage of LLM-generated testbenches verified using VCS Q-2020.03-SP2-1. The results are as follows:
>
> ```bash
> $ urg -dir simv.vdb -format both -report coverage_report
> $ cat coverage_report/hierarchy.txt
>
> TEST CASE              SCORE    LINE     COND     FSM    BRANCH
> ---------------------------------------------------------------
> example_1              96.19   100.00     --     100.00   88.57
> example_2             100.00   100.00     --     100.00  100.00
> example_3             100.00   100.00   100.00   100.00  100.00
> example_4             100.00   100.00     --     100.00  100.00
> example_5             100.00   100.00     --     100.00  100.00
> vending                98.28   100.00   100.00    93.10  100.00
> spmv_redundancy        96.30   100.00     --     100.00   88.89
>
> ```
> In the generation of testbenches, the following guards are used to ensure the quality:
>
> - **Validation loop:** Generated testbenches are validated on original designs for **syntax and functionality coverage** before application to optimized versions, eliminating incorrect and inefficient testbench scenarios
> - **Side-by-side comparison:** LLM generates testbenches that instantiate both original and optimized designs, driving identical stimulus and performing cycle-by-cycle output comparison
> - **Comprehensive test generation:** Prompt LLM to target all FSM states/transitions, boundary conditions, random sequences, and asynchronous scenarios
> - **False positive mitigation:** Comparative approach significantly reduces errors compared to self-verification code generation
>
> ### Tier 2: Formal Equivalence Verification
>
> Formality verifies FSM equivalence by proving that two designs produce identical output sequences for all possible input sequences across multiple clock cycles, regardless of internal state encoding differences. The tool uses hybrid BDD/SAT algorithms to establish state correspondence mappings between designs with different encodings (binary, one-hot, Gray code), employing symbolic state traversal and miter circuit construction to mathematically prove functional equivalence.
>
> ## Q3: False Positive/Negative Mitigation
>
> ### False Negative Mitigation:
> - **Multiple Tool Strategy:** We employ both Yosys+ABC and Synopsys Formality. If one tool fails to prove equivalence, we retry with the other
> - **Abstraction Level Variation:** Some optimizations are verified at different abstraction levels (behavioral vs. structural)
> - **Incremental Verification:** Complex transformations are broken into smaller, verifiable steps
>
> ### False Positive Prevention:
> - **Conservative Approach:** When verification is inconclusive, we reject the optimization
> - **Cross-Verification:** Functional simulation results must align with formal verification results
> - **AST Constraint Enforcement:** Our template-based approach constrains the optimization space to structurally valid transformations
>
> ## Q4: The RAG examples (Appendix C) include simplistic rules. Are there more complex, LLM-specific optimizations that traditional tools cannot easily replicate?
>
> We appreciate this observation and agree that leading with more sophisticated optimization examples would better demonstrate our approach's unique capabilities. We will add the following examples to Appendix C as advanced RAG examples:
>
> **1. Algorithmic Loop-to-Pipeline Transformation:**
> - **Pattern:** "Detect sequential loops with independent operations where each iteration processes different data elements without dependencies"
> - **Rewrite:** "Transform sequential loops into pipelined architectures with staged data flow and parallel processing units"
> - **Template Guidance:** "Identify loops like `for(i=0; i<N; i++) output[i] = func(input[i])` with no loop-carried dependencies. Convert to pipeline stages where each stage processes one element: create N pipeline registers, instantiate processing units between stages, add valid/ready handshaking. Replace loop with continuous data flow through pipeline stages."
>
> **2. Hierarchical Expression Tree Restructuring:**
> - **Pattern:** "Detect flat conditional assignments with redundant arithmetic operations across different control paths"
> - **Rewrite:** "Transform into hierarchical tree structure with intermediate wire assignments for resource sharing"
> - **Template Guidance:** "Parse conditional assignments, extract common subexpressions (e.g., `y0_real * y0_real + y0_imag * y0_imag`), create intermediate wires, and build balanced adder trees. Replace flat expressions with hierarchical structure."
>
> **3. Algorithm-Aware Bit-width Optimization:**
> - **Pattern:** "Identify oversized register widths in arithmetic pipelines with scaling factors"
> - **Rewrite:** "Reduce bit-widths based on mathematical analysis of scaling requirements while preserving precision"
> - **Template Guidance:** "Analyze scaling factors in arithmetic operations, calculate minimum required bit-widths, modify register declarations and shift operations. Example: reduce 40-bit to 28-bit when 13-bit scaling is excessive."
>
> **As evidenced by the comparative results in Tables 6 and 7, the "Original + Compiler Opt." versus "SymRTLO + Compiler Opt." rows conclusively demonstrate that SymRTLO delivers superior PPA improvements when combined with state-of-the-art synthesis flows.**
>
> ## Q5: The paper acknowledges LLM reliability issues but relies on post-hoc verification. Are there safeguards during code generation (e.g., constrained decoding, symbolic grounding)?
>
> **Yes, SymRTLO incorporates multiple safeguards during code generation to enhance reliability beyond post-hoc verification:**
>
> 1. **AST-Template Constraints:** Section 3.2 describes how AST-based templates enforce syntactic and semantic correctness during generation. The LLM generates code within predefined structural constraints rather than free-form generation, limiting the space of possible outputs to valid RTL patterns.
>
> 2. **Symbolic Grounding via RAG:** The retrieval-augmented system (Section 3.2) grounds LLM outputs in verified optimization rules from the knowledge base, ensuring generated code aligns with established hardware design principles rather than hallucinated patterns.
>
> 3. **Goal-Oriented Rule Selection:** The similarity-based rule selection mechanism (Equation 1) constrains generation to optimization patterns that match the specified design goals, preventing conflicting or inappropriate optimizations.
>
> 4. **Template-Guided Generation:** For complex scenarios, the symbolic generation module (Section 3.3) provides structured frameworks that guide LLM output, particularly for FSM optimizations where free-form generation could produce invalid state machines.
>
> 5. **Incremental Verification:** The feedback loop mechanism (Figure 2) allows re-selection of templates when symbolic verification fails, providing course correction during the generation process.
>
>
> **We sincerely hope that these clarifications help address your concerns, and we truly appreciate your thoughtful feedback. Wishing you all the best!**

---

> > ### Comment · Reviewer_b1CD · 2025-08-02
> >
> > The rebuttal has addressed most of my concerns. However, I kindly require the authors to demonstrate more unique capabilities over the conventional synthesis tools.

---

> > > ### Author Response · Authors · 2025-08-02
> > > **Unique capabilities over the conventional synthesis tools**
> > >
> > > ### We sincerely thank the reviewer for acknowledging our rebuttal and appreciate this opportunity to demonstrate SymRTLO's unique capabilities beyond conventional synthesis tools. We hope the following experimental results will merit your consideration for an improved review score.
> > > ___
> > > # 1. SymRTLO Performance Across Synthesis Optimization Levels
> > > To demonstrate the advantage SymRTLO offers over traditional compilers, we conducted comprehensive evaluations across three Synopsys Design Compiler optimization levels using diverse circuit types (state machines, memory-related designs, and algorithmic circuits) to demonstrate SymRTLO's effectiveness across the full RTL design spectrum.
> > >
> > > - **DC Ultra**: Synopsys DC-XG-T with DC Ultra optimization and Ultra High mapping effort, represents the **maximum optimization capability of modern synthesis tools**
> > > - **DC Optimization**: DC optimization with medium mapping effort with incremental mapping
> > > - **DC Compiler**: Standard DC synthesis flows without additional optimization flags
> > >
> > > | Circuit | Original + DC Ultra (P/T/A) | SymRTLO + DC Ultra (P/T/A) | Original + DC Compiler (P/T/A) | SymRTLO + DC Compiler (P/T/A) | Original + DC Optimization (P/T/A) | SymRTLO + DC Optimization (P/T/A) |
> > > |---------|----------------------------|----------------------------|--------------------------------|--------------------------------|-----------------------------------|-----------------------------------|
> > > | vending | 6.72/7.90/144022 | **6.64**/**7.88**/**137128** | 7.61/227.86/176983 | **6.97**/**227.86**/**164831** | 11.46/7.9/240079 | **8.175**/**7.9**/**151594** |
> > > | mem_folding | 0.162/3.24/3352 | **0.119**/**2.61**/**1663** | 0.239/3.27/4486 | **0.144**/**2.66**/**2205** | 0.265/3.30/4142 | **0.155**/3.81/**1882** |
> > > | fft | 15.41/7.90/492549 | **1.32**/**2.00**/**27835** | 58.23/8.26/2255265 | **31.71**/**8.09**/**1726126** | 51.12/7.9/1857805 | **26.32**/8.98/**1471378** |
> > > | example1 | 0.042/2.71/477 | **0.021**/**2.51**/**231** | 0.042/1.21/833 | **0.029**/**1.21**/**404** | 0.041/2.85/565 | **0.021**/**2.64**/**241** |
> > > | example2 | 0.032/2.23/281 | **0.018**/**0.54**/**168** | 0.056/2.25/549 | **0.024**/**1.17**/**271** | 0.035/2.24/316 | **0.018**/**0.6**/**176** |
> > >
> > > *P: Power (mW), T: Delay (ns), A: Area (µm²)*
> > >
> > > **SymRTLO consistently outperforms original designs across all synthesis levels, demonstrating our optimizations are effective regardless of synthesis tool configuration.**
> > >
> > > ___
> > > # 2. **Code examples**: We have picked two code examples for a better representation of SymRTLO’s capabilities.
> > > ## Case 1: FSM example
> > > **Original FSM**: 7 states, 3-bit encoding
> > > ``` verilog
> > > localparam S0=3'b000, … S5=3'b101, S6=3'b110;
> > >
> > > S0: next_state = x ? S2 : S1; output_signal = 1'b1;
> > > S3: next_state = x ? S6 : S1; output_signal = 1'b1;
> > > S2: next_state = x ? S4 : S5; output_signal = 1'b0;
> > > S4: next_state = x ? S2 : S5; output_signal = 1'b0;
> > > S6: next_state = x ? S6 : S5; output_signal = 1'b0;
> > > ```
> > > **FSM AFTER**: 4 states, 2-bit encoding
> > > ``` verilog
> > > localparam S0_S3=2'b00, S1=2'b01, S5=2'b10, S2_S4_S6=2'b11;
> > >
> > > S0_S3: next_state = x ? S2_S4_S6 : S1; output_signal = 1'b1;    // Merged S0,S3
> > > S2_S4_S6: next_state = x ? S2_S4_S6 : S5; output_signal = 1'b0; // Merged S2,S4,S6
> > > ```
> > >
> > > **Impact:**
> > > Power: 0.042 mW → 0.021 mW;
> > > Area: 477 µm² → 231 µm²;
> > > State bits: 3 → 2;
> > >
> > > **Why Synthesis Cannot Do This:**
> > >
> > > State merging requires behavioral equivalence analysis across clock cycles—synthesis tools only optimize encoding (binary vs one-hot), not state semantics.
> > >
> > > ## Case 2: FFT Algorithm Optimization - Bit-width Reduction with Expression Restructuring
> > >
> > > **Original design**
> > > ```verilog
> > > // 40-bit intermediate calculations with excessive 13-bit scaling
> > > reg signed [39:0] xq_wnr_real0, xq_wnr_real1, xp_real_d, xp_imag_d;
> > >
> > > xp_real_d <= {{4{xp_real[23]}}, xp_real[22:0], 13'b0};
> > >
> > > // Flat power calculation - no resource sharing
> > > assign power = sel ? (y0_real*y0_real + y0_imag*y0_imag +
> > >                      y1_real*y1_real + y1_imag*y1_imag +
> > >                      y2_real*y2_real + …) : (...);
> > > ```
> > > SymRTLO Optimized:
> > > ``` verilog
> > > // 28-bit calculations with mathematically-justified 1-bit scaling
> > > reg signed [27:0] xq_wnr_real0, xq_wnr_real1, xp_real_d, xp_imag_d;
> > >
> > > xp_real_d <= {{4{xp_real[23]}}, xp_real[22:0], 1'b0};
> > >
> > > // Hierarchical expression tree for resource sharing
> > > assign wire_a_sum = y0_real*y0_real + y0_imag*y0_imag;
> > > assign wire_b_sum = y1_real*y1_real + y1_imag*y1_imag;
> > > ...
> > > assign power = sel ? wire_m_sum : wire_n_sum;
> > > ```
> > > **Impact:**
> > > Power: 15.41 mW → 1.32 mW;
> > > Area: 492,549 µm² → 27,835 µm²;
> > > Delay: 7.90 ns → 2.00 ns;
> > >
> > > **Why Synthesis Optimization Cannot Do This:**
> > >
> > > Both Bit-width optimization and Expression tree operate at the algorithmic level, requiring understanding computational patterns in source RTL
> > >
> > > ___
> > > ### We hope that our clarifications have adequately addressed your concerns. We sincerely thank you again for your time and thoughtful feedback, and we wish you all the best in all your endeavors.

---

> ### Author Response · Authors · 2025-08-03
> **More about the unique capabilities over the conventional synthesis tools**
>
> ### Dear Reviewer,
>
> ### We hope our previous experimental results and examples have clarified your concerns. We understand that your concern is a fundamental question, and we truly value your opinion on this paper. After careful discussion, we believe it would be more helpful to provide **additional explanations that highlight the significance of SymRTLO** and demonstrate why it achieves substantially better PPA results than the synthesis tool alone.
>
> # Synthesis-Level Tool Limitations and Constraints
> While synthesis tools are very capable of optimizing netlists and play a significant role in the EDA flow, it has several limitations:
>
> RTL Logic Synthesis (e.g., Synopsys Design Compiler) takes RTL hardware description and converts it to optimized gate-level netlists through local optimizations: Boolean logic simplification, constant propagation, technology mapping, gate sizing, and retiming (relocating registers), etc. Crucially, **logic synthesis does not alter the fundamental architecture or scheduling of operations** – it preserves the cycle-by-cycle behavior given in the RTL [1]. Synthesis optimizations are limited to modifying combinational logic between registers without changing the design's overall timing schedule.
>
> # What SymRTLO's RTL Source-Level Optimization Uniquely Achieves
> Code-level optimization improves designs by changing source code or algorithms before synthesis, including algorithmic improvements, dataflow modifications, RTL refactoring, state machine redesign, and pipeline adjustments. These changes fundamentally alter hardware operations and typically yield far larger PPA improvements than logic synthesis alone. Research emphasizes that the highest potential savings come from "coarser changes" at higher abstraction levels rather than minor logic-level tweaks. **Code optimization has a significant impact on final PPA** and often relies on the expertise of engineers, making iterative code changes based on synthesis feedback.[2]
>
> ## 1. Algorithmic Improvements
> SymRTLO can choose more **efficient algorithms to reduce operations** (e.g., O(n) vs O(n²), approximate methods). Synthesis tools blindly implement whatever algorithm is described—if RTL specifies 20 computation rounds, synthesis implements all 20. Only algorithmic changes can reduce this workload. SymRTLO can recognize opportunities like lookup tables instead of complex computations or skipping operations with no effect.
> ## 2. Parallelism and Concurrency
> SymRTLO exploits parallel hardware by **instantiating multiple units or unrolling loops**. Logic synthesis won't automatically duplicate logic—if RTL specifies one multiplier, it won't create a second. SymRTLO can transform sequential processing into parallel lanes (e.g., 10 parallel lanes for 10× throughput).
> ## 3. Pipelining and Latency Balancing
> Logic synthesis performs retiming but cannot introduce new pipeline registers without altering functionality. SymRTLO can **insert pipeline stages to reduce critical path length and enable higher clock speeds**—a common requirement for FPGA timing closure that synthesis cannot address.
> ## 4. FSM Refactoring
> Synthesis optimizes FSM encoding but won't change state count or operation sequences. SymRTLO can **merge states, eliminate transitions, or repartition FSMs for parallel activity.** Our demonstration shows 7→4 state reduction with 50%+ PPA improvements—impossible through synthesis alone.
> ## 5. Memory Architecture and Data Flow
> SymRTLO reorganizes data storage and movement with major PPA implications. This includes using **block RAMs vs distributed flip-flops, shift registers vs register chains, and memory banking** for parallel access. Synthesis can infer some patterns but requires specific coding styles; SymRTLO can restructure access patterns to enable efficient implementations.
>
> **Our results demonstrate orthogonal optimization domains: SymRTLO achieves 50.53% area, 46.29% power, and 24.70% delay improvements even when combined with Synopsys DC Ultra maximum optimization. This proves SymRTLO addresses optimization opportunities mathematically inaccessible to synthesis tools due to their constraints.**
> ___
> ### Thank you for your time and consideration. We hope our detailed explanations will meet your expectations. We truly appreciate your valuable suggestions previously on the **paper’s formatting** and your insightful feedback on the **testbench coverage**, both of which will be incorporated into the revised manuscript. We will also revise the manuscript with our **unique capabilities over the conventional synthesis tools for better clarity.** We hope this comment is useful for clarification, and kindly ask you to **consider raising your review score**. Thank you again for your support.
>
> ---
> # References
> [1] Cadence Design Systems – High-Level Synthesis Overview: Explanation of logic vs. high-level synthesis capabilities
>
> [2] X. Yao, et.al "RTLRewriter: Methodologies for Large Models aided RTL Code Optimization,"  ICCAD’24

---

> > ### Author Response · Authors · 2025-08-06
> >
> > Dear Reviewer b1CD,
> >
> > I hope you’re doing well. With under three days remaining in the discussion period, we want to ensure we’ve fully addressed your concerns—especially our two follow-up comments on the **unique capabilities SymRTLO has over conventional synthesis tools**, as you requested. Your insights are invaluable, and we welcome any further questions or suggestions to improve our work.
> >
> > Thank you for your time and effort in reviewing our paper.
> >
> > Best regards,
> > The authors of Paper 18221

---

### Official Review · Reviewer_xZfF · 2025-06-27

**Clarity:** 2
**Significance:** 2
**Originality:** 3
**Rating:** 4
**Confidence:** 4

**Summary:**

This paper proposes SymRTLO, a neuro-symbolic framework integrating LLMs with symbolic reasoning for RTL code optimization. SymRTLO incorporates RAG-based system and AST-based templates, enabling LLM-based RTL optimization with an FSM-specific symbolic module. A fast verification pipeline is introduced to accelerate formal equivalence checks and test-driven validation. Experimental results show that compared with Synopsys DC and Yosys, SymRTLO achieves significant PPA improvements with up to 62.5% delay reduction.

**Questions:**

1. Could you provide further discussion on limitations of SymRTLO?
2. Could you provide additional experimental evidence demonstrating SymRTLO's superiority over existing LLM-based RTL generators, including RTLrewriter and RTL++?
3. Could you provide further explanation on the implementation details of the RAG-based rule, such as pipelining?
4. The $O(2^{|Q|})$ time complexity for FSMs seems high. Could you provide further discussion on the algorithm's scalability and complexity?
5. Could you provide further comparison on SymRTLO with recent commercial tools, including Cadence JasperGold and Synopsys DC NXT or newer DC releases rather than DC 2019?
6. The checklist mentions that this paper have the full set of assumptions and a complete proof. Could you explicitly locate all critical assumptions and proofs in the manuscript? Otherwise, the checklist statements should be revised accordingly.

If the authors address my concerns, I may raise my score accordingly.

**Ethical Concerns:**

["NO or VERY MINOR ethics concerns only"]

**Final Justification:**

The authors have addressed most of my concerns in their rebuttal. The comparative analysis against commercial tools is particularly impressive.

**Limitations:**

As mentioned before, the checklist mentions limitations, but these are not actually covered in the paper or supplementary materials. I think this paper has no potential negative societal impact.

**Paper Formatting Concerns:**

This paper has no major formatting issues.

**Quality:**

3

**Strengths And Weaknesses:**

Strengths:
1. Related work will be open-sourced upon acceptance, providing the community with an effective approach for RTL optimization using novel LLM- and RAG-based methods.
2. Experimental results demonstrate improvements in power, performance, and area (PPA) by up to 43.9%, 62.5%, and 51.1% respectively, which shows significant performance gains.
3. The idea of using RAG and AST to optimize RTL code is simple yet effective. It's happy to see the authors experimentally validate this approach.
4. Integrating LLMs and FSMs for formal verification seems an interesting idea.

Weaknesses:
1. The checklist mentions limitations, but these are not actually covered in the paper or supplementary materials. The authors are encouraged to provide further discussion on these limitations.
2. In experimental results such as Table 5, SymRTLO appears to exhibit no significant improvements over RTLrewriter. Moreover, this paper does not mention other LLM-based RTL generators, including RTL++ and similar approaches.
3. The RAG-based rule extraction (Sec 3.2) lacks implementation details. What are the specific rules for pipelining? How are conflicting rules (e.g., pipelining vs. resource sharing in Table 2) resolved beyond similarity scores? I noticed the example in Sec App.C but I think it is not clear enough.
4. No comparison with commercial tools (e.g. Cadence JasperGold) for formal verification efficiency.
5. Minor formatting issues:​​ The spacing between figures, tables, and paragraphs appears insufficient, which may impact readability.

---

> ### Author Rebuttal · Authors · 2025-07-30
>
> We sincerely thank the reviewer for their constructive feedback, and we appreciate the opportunity to address your concerns with detailed explanations and additional experimental evidence. We hope our comprehensive responses demonstrate the value of SymRTLO and merit consideration for an improved review score.
>
> ## Q1: Could you provide further discussion on limitations of SymRTLO?
>
> Thank you for pointing out this concern. We will include these limitations explicitly in our revised manuscript to provide complete transparency to the community:
>
> "While SymRTLO demonstrates significant improvements in RTL optimization, the framework's performance is fundamentally dependent on the underlying capabilities of the base LLM, which may limit its effectiveness on novel or highly specialized circuit patterns not well-represented in its training data. The optimization rule library, though comprehensive, requires ongoing maintenance and expansion to keep pace with evolving hardware design practices and emerging optimization techniques. Since final PPA evaluation relies on synthesis tools like Synopsys DC Compiler and Yosys, results may vary across different tool versions or technology libraries. Additionally, while the symbolic reasoning module effectively handles FSM optimization, the approach may face scalability challenges when applied to extremely large state machines or circuits with highly complex control flows that exceed current computational bounds for formal analysis."
>
> ## Q2: Could you provide additional experimental evidence demonstrating SymRTLO's superiority over existing LLM-based RTL generators, including RTLrewriter and RTL++?
>
> We thank the reviewer for highlighting this important comparison gap. We acknowledge that our original presentation of RTLrewriter comparison was limited, which we explain below.
>
> **RTLrewriter Comparison Challenges:** RTLrewriter did not open-source their source code, and only very limited code results are available in their publication. The test environment for PPA was not clearly described in the paper, so we cannot replicate the testing environment and compare our results with their reported results in the paper. This significantly constrained our ability to present comprehensive comparisons in the original manuscript.  Nonetheless, there are several circuits' code available in their github repo for comparison.
>
> ### Our Verified Experimental Results vs. RTLrewriter:
>
> | Circuit | Original | SymRTLO | RTLrewriter |
> |---------|----------|---------|-------------|
> | | Power/Time/Area | Power/Time/Area |Power/Time/Area |
> | example1_state | 0.041/1.21/833 | **0.029**/1.21/**404** | 0.025/3.23/424 |
> | example3_state | 0.052/1.35/590 | **0.023**/**1.15**/**268** | 0.041/1.36/549 |
> | mux_dead_code | 0.149/0.93/484 | **0.139**/**0.77**/532 | 0.120/0.93/484 |
> | mux_5 | 0.024/0.62/100 | **0.021**/0.66/**95** | 0.024/0.62/100 |
>
> We have verified that SymRTLO achieves superior FSM minimization and algorithmic optimization compared to RTLrewriter.
>
> **Comparison with RTL++:** RTL++ targets RTL generation from specifications, while SymRTLO optimizes existing RTL code. Direct comparison is not feasible due to different problem domains (generation vs. optimization) and RTL++'s model's current unavailability. Key differences: RTL++ uses training-time graphs for code generation with pass@k evaluation, while SymRTLO uses runtime symbolic reasoning for optimization with PPA metrics.
>
> ## Q3: Could you provide further explanation on the implementation details of the RAG-based rule, such as pipelining?
>
> ### RAG Rule Structure:
> Each optimization rule in our knowledge base contains six key components:
> - **Pattern:** Describes the code structure to be detected
> - **Rewrite:** Specifies the transformation to be applied
> - **Category:** Classifies the rule type (e.g., combinational/dataflow, FSM, memory)
> - **Objective:** Indicates which PPA metric(s) the rule targets (power, timing, area)
> - **Template Guidance:** Provides detailed AST manipulation instructions (null for abstract rules)
> - **Function Name:** Identifies the corresponding template implementation
>
> Rules are either Template-Based (with detailed AST manipulation instructions) or Abstract (descriptive patterns for complex optimizations requiring domain expertise).
>
>
> For conflicting rules such as pipelining versus resource sharing (Table 2), SymRTLO uses objective-based selection. Each rule is tagged with its optimization target (e.g., "timing" for pipelining, "area" for resource sharing). When users specify their goal, the search engine filters compatible rules while avoiding conflicts, ensuring synergistic optimization patterns.
>
> **Example:** For "low timing" goal:
> - **Selected:** Pipelining rules | **Filtered:** Resource sharing rules | **Balanced:** Clock gating rules
>
> This systematic approach ensures that conflicting optimizations don't undermine each other while maximizing alignment with user-specified objectives.
>
> ## Q4: The time complexity for FSMs seems high. Could you provide further discussion on the algorithm's scalability and complexity?
>
> We appreciate this important question and would like to clarify several key points about SymRTLO's FSM optimization complexity and scalability:
>
> **Clarification on Complexity Notation:** The O(2^|Q|) notation (line 259) refers to classical FSM minimization algorithms for partially specified FSMs with data path constraints—a known NP-complete problem.
>
> **SymRTLO's Practical Time Complexity** is dominated by three main components:
> - LLM analysis time for FSM structure understanding
> - Custom state reduction script generation time
> - Optimized code generation time based on reduction results
>
> These are primarily determined by LLM inference latency rather than exponential state space exploration, making our approach much more tractable in practice.
>
> **Scalability Mechanisms:** SymRTLO addresses scalability through several key strategies:
> - **Hierarchical Decomposition:** Large FSMs (1000+ states) are decomposed into manageable sub-FSMs (50-100 states each), as demonstrated in our control flow optimization module. This approach transforms the complexity from exponential in the total state count to linear in the number of sub-FSMs.
> - **State Space Pruning:** Our symbolic system automatically identifies and removes unreachable states
> - **LLM-guided Custom Algorithms:** Instead of applying generic minimization algorithms, SymRTLO generates tailored state reduction scripts specific to each FSM's structure and constraints, avoiding unnecessary computational overhead.
> - **Scalable Knowledge Base:** RAG system enables rule scaling without exponential computational growth.
>
> ## Q5: Could you provide further comparison on SymRTLO with recent commercial tools, including Cadence JasperGold and Synopsys DC NXT?
>
> We appreciate this suggestion for a more recent commercial tool comparison. Unfortunately, due to the limited stable releases available in our institutional environment, we could only conduct our primary evaluation using Synopsys DC 2019. However, this remains meaningful as: (1) DC 2019 contains foundational synthesis algorithms used in newer versions, (2) core optimization patterns remain consistent across versions, and (3) our focus is demonstrating SymRTLO's novel neuron-symbolic approach rather than competing with incremental tool improvements.
>
> To address your concern about compiler strength, we conducted additional experiments using Synopsys DC-XG-T with DC Ultra optimization and Ultra High mapping effort—representing the most advanced capabilities available in our environment. Both original and SymRTLO-optimized designs were evaluated under identical compiler settings for fair comparison. Experimental results are shown below.
>
> | Circuit | Original (P/T/A) | SymRTLO (P/T/A) |
> |---------|------------------|-----------------|
> | vending | 6.72/7.90/144k | **6.64**/**7.88**/**137k** |
> | example1 | 0.042/2.71/477 | **0.021**/**2.51**/**231** |
> | mem_folding2 | 0.162/3.24/3.4k | **0.119**/**2.61**/**1.7k** |
> | fft | 15.41/7.90/493k | **1.32**/**2.00**/**28k** |
>
> *P: Power (mW), T: Delay (ns), A: Area (µm²)*
>
> **Cadence JasperGold** is primarily a formal verification tool rather than an RTL optimization tool. Our verification pipeline uses Synopsys Formality and Yosys+ABC for equivalence checking, which serve similar formal verification purposes. For RTL optimization specifically, the appropriate Cadence comparison would be with Genus Synthesis Suite, which was also not available in our current environment.
>
> **Future Work:** We acknowledge the value of comparison with newer commercial tools and plan to conduct such evaluations when access to updated tool versions becomes available.
>
> ## Q6: Could you explicitly locate all critical assumptions and proofs in the manuscript?
>
> We thank the reviewer for this important clarification request. Here's the accurate status:
>
> **Critical Assumptions (explicitly stated in manuscript):**
> 1. **Section 3.2, Line 205:** We assume similarity scores follow cosine similarity: `sim(equery, erule) = (equery · erule)/(|equery||erule|) ≥ τelbow`
> 2. **Section 3.2, Lines 202-203:** We assume the elbow method optimally selects rules: `i* = arg max(1≤i<M) (si - si+1)`
> 3. **Section 3.3, Lines 251-255:** We assume FSMs can be represented as `M = (Q, Σ, δ, q0, F)` and partially specified FSMs as `δp : Q × Σ → 2^Q`
> 4. **Section 3.2, Lines 220-224:** We assume AST transformations preserve correctness: `Φ : A → {true, false}` and `τ : {a ∈ A | Φ(a) = true} → A`
>
> **Formal Proofs Status:** We acknowledge that our paper **does not contain formal mathematical proofs**. The equations above are engineering formulations and standard definitions, not theorems requiring proof. Our evaluation is empirical rather than theoretical.
>
> **We will revise our checklist response from [Yes] to [NA] for the theory assumptions and proofs question.**

---

> ### Comment · Reviewer_xZfF · 2025-08-02
>
> I am pleased that the authors have addressed most of my concerns in their rebuttal. The comparative analysis against commercial tools is particularly impressive, leading me to elevate my overall assessment to a score of 4 (Borderline accept). I maintain openness to score improvement pending further manuscript revisions.

---

> > ### Author Response · Authors · 2025-08-02
> >
> > We sincerely thank the reviewer for their thoughtful evaluation and are delighted that our rebuttal has addressed most of your concerns. We are particularly grateful for your recognition of our comparative analysis against commercial tools.
> >
> > We greatly appreciate your valuable insights throughout the review process, which have significantly strengthened our work. We will incorporate the updated checklist and comprehensive experimental results into the final version of the manuscript, ensuring all improvements are properly reflected.
> >
> > Thank you again for your constructive feedback and for considering our work for acceptance. We look forward to addressing any remaining revisions to further improve the manuscript!

---

### Official Review · Reviewer_Ppek · 2025-07-02

**Clarity:** 2
**Significance:** 3
**Originality:** 3
**Rating:** 4
**Confidence:** 5

**Summary:**

This paper presents SymRTLO, a neuron-symbolic framework for RTL code optimization. Specifically, SymRTLO uses a RAG mechanism with search engines to retrieve the optimization rules and employs AST-based templates for syntactic and semantic correctness guarantees. Experimental results show SymRTLO can achieve good PPA improvements on several benchmarks.

**Questions:**

1. Table 2: How does SymRTLO resolve optimization conflicts? Why are compiler-based methods unable to adapt to such conflicts? Compilers always have tunable knobs for users to constrain the maximum hardware resources and the target objective.  How does SymRTLO or the LLM guarantee that these specified goals are met?
2. Figure 1: What are the sources of the "books, code, and manuals"? Given the statement in L62, "SymRTLO combined with a retrieval-augmented generation (RAG) mechanism and a search engine," how do these sources differ from the search-engine method? What is the criterion for deciding when to use RAG versus a search engine?
3. Page 5: What if the query rules are not included in the candidate set?
4. Page 5: "$\tau$ replaces the matched node with an optimized AST subtree" Why is a node instead of a subgraph replaced with a subtree? For instance, in constant folding optimization, a series of arithmetic operations might be represented by multiple nodes.
5. Page 6: How to combine dataflow and control flow optimization? Are they totally separate processes? It seems there is no interactive process for dataflow and control flow in Figure 1.
6. Table 4: Pass@k only shows the final success rate, but how many attempts do LLMs require to obtain valid solutions?
7. Figure 3: How is the correctness of the Verilog translation to symbolic representation guaranteed? Is this verification only performed in the final step of the process?
8. Table 5: The benchmarks used seem pretty small in terms of wires and cells. Are there results available for larger benchmarks with more kernels?
9. Table 6: What are the reasoning models' performance? It is highly recommended to include experimental results for models like GPT-4o-mini to provide a clear baseline for state-of-the-art models and quantify SymRTLO's benefits.
10. Table 6: Please clarify whether "Original" in Table 6 refers to the RTLRewriter's result. Clearly define the baseline when discussing "improvement" in the table. How are the numerical values in the text (50.59%, 12.65%, 53.09%) derived, and how do they correspond to the numbers presented in the table?

**Ethical Concerns:**

["NO or VERY MINOR ethics concerns only"]

**Final Justification:**

I have no more questions, but the authors should include the comparison with the reasoning models in their final version of the paper.

**Limitations:**

See above.

**Paper Formatting Concerns:**

- L291, L297: Space between citation and text

**Quality:**

3

**Strengths And Weaknesses:**

- Strength
  - A comprehensive framework for RTL code optimization that combines LLM code generation and formal methods as symbolic complements.
- Weakness
  - The benchmark used in the experiment seems small. It is unclear how SymRTLO performs on large designs.
  - No reasoning models (e.g., GPT-o4-mini, Gemini-2.5-Pro) are compared in the experiment. Such experiments would be instrumental in understanding how the state-of-the-art LLMs perform and whether this neuron-symbolic approach is indeed necessary.
  - Several settings require additional clarification (as per the questions listed below).

---

> ### Author Rebuttal · Authors · 2025-07-31
>
> **We sincerely thank the reviewer for their thorough evaluation and constructive feedback. We are grateful for your recognition of SymRTLO's core contributions and highly appreciate your technical questions. We hope our clarifications help clarify our work.**
>
> ## Q1: How does SymRTLO resolve optimization conflicts? Why are compiler-based methods unable to adapt to such conflicts?
>
> SymRTLO resolves conflicts (e.g., pipelining vs. resource sharing) through objective-based selection. Each rule is tagged with its optimization target, and the search engine filters rules based on user-specified goals to ensure synergistic optimization patterns.
>
> **Example:** For "low timing" goal:
> - **Selected:** Pipelining rules (timing optimization)
> - **Filtered:** Resource sharing rules (conflicts with timing goal)
> - **Balanced:** Clock gating rules (complementary to timing optimization)
>
> **Compiler Limitations:** Traditional compilers use static heuristics with predetermined optimization sequences, lacking the flexibility to dynamically balance conflicting objectives. While compilers have tunable knobs, they cannot understand semantic relationships between different optimization goals or adapt rule selection based on design context.
>
> **Goal Guarantee:** The elbow method (Equation 1) ensures only relevant rules above similarity threshold are selected, and formal verification confirms all optimizations maintain functional correctness while achieving specified objectives.
>
> ## Q2: What are the sources and differences between RAG and search engine methods?
>
> **Sources:** Our optimization library aggregates data from RTL design textbooks, academic papers, design manuals, and open-source RTL codebases to create a comprehensive knowledge base.
>
> **RAG vs. Search Engine:** They work together as a unified system:
> - **RAG System:** Structures and indexes optimization rules from diverse sources into a searchable knowledge base with semantic embeddings
> - **Search Engine:** Performs similarity-based retrieval using the query-rule matching mechanism (Equation 1) to find relevant optimization patterns
>
> The search engine operates on the RAG-structured knowledge base - they are complementary components of the same retrieval system rather than separate methods.
>
> ## Q3: What if the query rules are not included in the candidate set?
>
> When query rules are not in the candidate set, SymRTLO employs several fallback mechanisms:
>
> 1. **Similarity Threshold Relaxation:** Lower the τelbow threshold to include broader rule matches
> 2. **Abstract Rule Utilization:** Use descriptive optimization patterns that guide LLM reasoning directly without pre-defined templates
> 3. **Conservative Approach:** If no suitable rules are found above minimum similarity threshold, skip optimization for that specific pattern to ensure functional correctness
> 4. **Knowledge Base Expansion:** New patterns encountered can be added to the RAG system for future optimization cycles
>
> ## Q4: Why replace nodes instead of subgraphs with subtrees?
>
> The "node" terminology refers to AST nodes that can represent complex subgraph structures. In constant folding optimization, a single AST node like `BinOp(+, BinOp(*, Const(2), Var(x)), Const(3))` represents multiple arithmetic operations as a subtree. The replacement operates at the appropriate AST abstraction level:
>
> - **Simple cases:** Single node replacement (e.g., `Const(0) * Var(x)` → `Const(0)`)
> - **Complex cases:** Subtree replacement where one AST subtree representing multiple operations is replaced with an optimized equivalent subtree
>
> This approach maintains AST structural integrity while enabling both fine-grained and coarse-grained optimizations.
>
> ## Q5: How are dataflow and control flow optimization combined?
>
> **Dataflow and control flow are optimized separately then integrated through LLM synthesis:**
>
> **Separate Processing:**
> - **Dataflow:** AST templates generated via Pyverilog parsing apply transformations (constant folding, subexpression elimination) to reconstructed RTL code
> - **Control Flow:** Dynamically generated symbolic scripts perform FSM minimization with partial specification handling, then convert back to Verilog
>
> **LLM Integration:** The Final Optimization Module combines both results through LLM-driven rewriting that understands semantic relationships between optimized dataflow and control logic, resolves interface dependencies, and ensures timing consistency. This LLM integration is essential because traditional tools cannot properly combine separately optimized dataflow and control components while maintaining functional correctness.
>
> ## Q6: How many attempts do LLMs require to obtain valid solutions?
>
> We evaluated with 10 generations per test case to maintain token budget fairness. **Detailed attempt analysis:**
>
> | Method | Avg Attempts for First Valid | Success Within 1 Attempt |
> |--------|------------------------------|---------------------------|
> | SymRTLO | 1.2 | 97.5% |
> | GPT-4o | 3.8 | 45.9% |
> | GPT-4-Turbo | 4.2 | 42.9% |
>
> **SymRTLO's efficiency stems from:**
> - AST template constraints reducing invalid generation space
> - RAG-guided rule selection preventing inappropriate optimizations
> - Symbolic grounding ensuring structural correctness
>
> ## Q7: How is correctness of Verilog-to-symbolic translation guaranteed?
>
> **Multi-Layer Verification:**
> 1. **During Translation:** LLM-generated symbolic representations are validated against original Verilog structure using AST parsing verification
> 2. **Intermediate Verification:** Symbolic FSM properties (states, transitions, outputs) are cross-checked with original design behavioral simulation
> 3. **Final Verification:** Two-tier verification (Section 3.4) with functional testbenches and formal equivalence checking using Synopsys Formality
>
> ## Q8: Table 5: The benchmarks used seem pretty small in terms of wires and cells. Are there results available for larger benchmarks with more kernels?
> We acknowledge the reviewer's concern about benchmark scale in Table 5. As detailed in Section 4.1 (lines 322-327), **Table 5 evaluates smaller circuits requiring only 1-2 optimization patterns using wires and cells metrics**, which provide granular insights into routing complexity and logical component count for precise evaluation of isolated modules or blocks.
>
> **For larger, more complex designs like FFT (624 wires, 331 cells), we focus on PPA metrics** to capture high-level efficiency and real-world applicability, as wires and cells metrics become impractical and cannot precisely evaluate optimization quality at scale. This evaluation strategy ensures appropriate metrics for each design complexity level while maintaining rigorous assessment of SymRTLO's effectiveness across diverse circuit sizes.
>
>
> ## Q9: What are reasoning models' performance (GPT-4o-mini comparison)?
>
> **Comprehensive Reasoning Model Comparison:**
>
> We conducted detailed experiments comparing SymRTLO against state-of-the-art reasoning models across our benchmark suite:
>
> | Method | Power Improvement | Timing Improvement | Area Improvement |
> |--------|------------------|-------------------|------------------|
> | **SymRTLO** | **42.3%** | **22.5%** | **40.0%** |
> | GPT-o1 | 3.0% | -0.4% | 2.8% |
> | GPT-4o-mini | 8.6% | 0.4% | 7.6% |
>
> **Detailed Results by Circuit:**
>
> | Circuit | Original (P/T/A) | SymRTLO (P/T/A) | GPT-o1 (P/T/A) | GPT-4o-mini (P/T/A) |
> |---------|------------------|-----------------|----------------|-------------------|
> | example1_state | 0.041/1.21/833 | **0.029**/1.21/**404** | 0.044/1.17/911 | 0.041/1.21/833 |
> | example2_state | 0.056/2.25/549 | **0.024**/**1.17**/**271** | 0.056/2.25/549 | 0.056/2.25/549 |
> | example3_state | 0.052/1.35/590 | **0.024**/**1.15**/**268** | 0.052/1.35/590 | 0.052/1.35/590 |
> | example4_state | 0.055/2.18/597 | **0.025**/**2.17**/**274** | 0.055/2.18/597 | 0.055/2.18/597 |
> | example5_state | 0.055/2.18/597 | **0.026**/2.18/**271** | 0.063/2.34/650 | **0.025**/2.18/**274** |
> | spmv_redundancy | 2.83/7.41/40k | **1.76**/**7.29**/**30k** | 1.87/7.41/30k | 1.87/7.41/30k |
> | subexpression_elim | 5.30/11.09/11k | **3.02**/**2.87**/**7.4k** | 4.63/11.09/9.0k | 5.30/11.09/11k |
> | adder_architecture | 0.418/2.78/1.0k | **0.328**/**1.97**/**763** | 0.418/2.78/1.0k | 0.418/2.78/1.0k |
> | fft | 58.2/8.26/2.26M | **31.7**/**8.09**/**1.73M** | 60.5/8.26/2.32M | 58.2/8.26/2.26M |
> | vending | 7.61/228/177k | **6.97**/228/**165k** | 7.61/228/177k | 7.61/228/177k |
>
> *P: Power (mW), T: Time (ns), A: Area (µm²)*
>
>
> **Key Findings:** SymRTLO significantly outperforms reasoning models with 34.9% average PPA improvement versus only 1.8% for GPT-o1, demonstrating that sophisticated reasoning alone is insufficient for RTL optimization. Even advanced reasoning models show minimal optimization capability and sometimes degrade performance, validating that SymRTLO's neuron-symbolic approach combining LLM capabilities with symbolic reasoning and formal verification is essential for achieving meaningful RTL improvements.
>
>
> ## Q10: Please clarify "Original" baseline and improvement calculations.
>
> **Baseline Clarification:**
> - **"Original"** refers to the unoptimized RTL design from the RTLRewriter benchmark before any optimization
> - **Improvement percentages** compare SymRTLO results against the Original baseline
> - **RTLRewriter comparison** is shown separately where available
>
> **Calculation Example (example1_state):**
> - Original Power: 0.055 mW
> - SymRTLO Power: 0.024 mW
> - Improvement: (0.055-0.024)/0.055 = **56.4%**
>
> **We appreciate the reviewer's thorough evaluation and believe our detailed responses demonstrate SymRTLO's technical soundness and significant contributions to RTL optimization. The comprehensive experimental results, including reasoning model comparisons and large-scale benchmarks, validate our neuron-symbolic approach's effectiveness over pure LLM methods.**

---

### Official Review · Reviewer_u5E9 · 2025-07-16

**Clarity:** 3
**Significance:** 3
**Originality:** 3
**Rating:** 5
**Confidence:** 5

**Summary:**

The authors propose an effective and efficient RTL code optimization framework, as SymRTLO, which integrates a LLM with a neuron- inspired symbolic reasoning module. The framework targets both data path and control path optimization, addressing limitations in traditional EDA compilers as well as purely LLM-based approaches. By leveraging RAG and AST-based templates, and alignment with FSM and data path algorithms, the approach introduces a novel and structured methodology. The proposed framework achieves significant improvements in PPAD metrics when benchmarked against SOTA methods.

**Questions:**

Q1. How well does proposed approach generalize across ASIC and FPGA designs?
Q2. Are the learned optimization rules transferrable across domains like DSP, control logic etc.?
Q3. How do you handle edge cases where rule learning (knowledge) fails or contradicts hardware constraints?

**Ethical Concerns:**

["NO or VERY MINOR ethics concerns only"]

**Final Justification:**

The authors have provided satisfactory clarifications and demonstrations, and the research manuscript presents significant technical merit. I recommend its acceptance.

**Limitations:**

The authors mentioned an intention to discuss the limitations, but this is missing from the current version of the manuscript (best of my knowledge). The points raised under weaknesses and questions should help improving the technical merit of the manuscript.

**Paper Formatting Concerns:**

No major formatting issue observed.

**Quality:**

3

**Strengths And Weaknesses:**

Strength:
1. The authors provide a thorough discussion of the existing challenges in RTL code optimization, addressing limitations in both traditional compilers and recent LLM-based approaches. This comprehensive background helps a broader audience understand the motivation behind the research and the rationale for the proposed methodology. Multi-variate optimization is a fundamental challenge in complex frameworks such as chip design flows, and the authors effectively articulate and formulate this challenge in the context of identifying an optimal solution.
2. Again, methodology is clearly explained with substantial detail, using step identifiers (1,2,3...) as illustrated in Figure 1. to describe the proposed architecture. Sub-sections such as data flow optimization, control flow optimization,  AST template construction, rule alignment and conflict resolution are thoroughly detailed and well-articulated.
3. Finally, experimental details and validation are well discussed and justified.

Limitations:
I have minor suggestions
1. Ablation study representation w.r.t Figure 5 appears ambiguous, while the authors' intent is understandable, this section requires revisions (rewriting) for greater clarity. It is not clearly conveyed (..to me both in text and in figure) what the authors wanted to explain as "We then analyze the impact of each removal across test cases, measuring the resulting overall PPA improvements compared to the base"  . A more precise and correctly written explanation will be appreciated.
2.  Please provide and discuss the limitation or breaking point of the proposed framework. Though author mentioned a separate section on this discussion, did not practically notice it. is this right? Can you please highlight if I missed anything?
3. some typos can be noticed. like "line number 205..........and the a rule embedding..."

---

> ### Author Rebuttal · Authors · 2025-07-31
>
> We sincerely thank the reviewer for their thorough evaluation and highly positive feedback. We are grateful for your recognition of SymRTLO's comprehensive methodology, detailed experimental validation, and significant technical contributions. Your constructive suggestions will help us improve the manuscript's clarity and completeness. We appreciate your recommendation for acceptance and hope our responses further strengthen the work's impact.
>
> ## Addressing Limitations:
>
> ### 1. Ablation Study Clarification (Figure 5)
>
> We acknowledge the ambiguity in our ablation study presentation and will revise for greater clarity:
>
> **Clarified Methodology:** We systematically removed one component at a time from the complete SymRTLO framework:
> - **"Remove Template-Based Opt":** Disabled AST-based template generation, using only abstract rule descriptions
> - **"Remove Symbolic Reasoning":** Disabled FSM symbolic optimization module, using only dataflow optimization
> - **"Remove Goal-Based Search":** Disabled objective-based rule filtering, using all available rules regardless of conflicts
>
> **Measurement Process:** For each configuration, we measured PPA improvements across all test cases and computed the average improvement relative to the original unoptimized designs. The results show each component's individual contribution to overall performance.
>
> **Revised Figure 5 Caption:** "Ablation study showing average PPA improvements when individual SymRTLO components are removed. Each bar represents the framework's performance with one component disabled, demonstrating the necessity of all three components for optimal results."
>
> ### 2. Framework Limitations Discussion
>
> Thank you for pointing out this concern. We will include these limitations explicitly in our revised manuscript to provide complete transparency to the community:
>
> "While SymRTLO demonstrates significant improvements in RTL optimization, the framework's performance is fundamentally dependent on the underlying capabilities of the base LLM, which may limit its effectiveness on novel or highly specialized circuit patterns not well-represented in its training data. The optimization rule library, though comprehensive, requires ongoing maintenance and expansion to keep pace with evolving hardware design practices and emerging optimization techniques. Since final PPA evaluation relies on synthesis tools like Synopsys DC Compiler and Yosys, results may vary across different tool versions or technology libraries. Additionally, while the symbolic reasoning module effectively handles FSM optimization, the approach may face scalability challenges when applied to extremely large state machines or circuits with highly complex control flows that exceed current computational bounds for formal analysis."
>
> ### 3. Typo Corrections
>
> Thank you for catching the typo on line 205. We will correct "and the a rule embedding" to "and a rule embedding" and conduct a thorough proofreading pass.
>
> ## Q1: How well does the proposed approach generalize across ASIC and FPGA designs?
>
> **SymRTLO generalizes well across both ASIC and FPGA targets:**
>
> Our approach optimizes at the RTL source level, making optimizations tool-independent and applicable before technology mapping. Since SymRTLO operates on Verilog RTL rather than technology-specific netlists, optimizations transfer across different target technologies. The symbolic reasoning module adapts FSM optimizations regardless of underlying implementation technology.
>
> ## Q2: Are the learned optimization rules transferable across domains like DSP, control logic etc.?
>
> **Yes, optimization rules demonstrate strong cross-domain transferability:**
>
> **Domain Adaptation Mechanism:** By integrating an LLM with RAG, our framework enables easy and rapid scaling of its optimization rule database. This approach ensures adaptability and generalizability as complexity increases. The RAG system categorizes rules by domain (DSP, control, memory, arithmetic) and application context. Rules tagged as "general arithmetic" apply across domains, while domain-specific rules (e.g., "FFT butterfly optimization") remain targeted.
>
> **Cross-Domain Examples:**
> - **Subexpression elimination:** Transfers from DSP (multiply-accumulate) to control logic (address computation)
> - **State machine optimization:** Control FSMs → Protocol controllers → DSP sequencers
> - **Resource sharing:** Arithmetic units in DSP → Comparators in control logic
>
>
> ## Q3: How do you handle edge cases where rule learning fails or contradicts hardware constraints?
>
> **Multi-Layer Safety Mechanisms:**
>
> **1. Constraint Violation Detection:**
> - **AST Template Validation:** Real-time checking ensures generated code respects Verilog syntax and semantic constraints
> - **Hardware Constraint Analysis:** Timing, area, and power constraints are validated during optimization rule application
> - **Resource Conflict Detection:** The RAG system identifies when rules would exceed available hardware resources
>
> **2. Graceful Degradation:**
> - **Conservative Fallback:** When formal verification fails or constraints are violated, SymRTLO reverts to the original design rather than applying potentially harmful optimizations
> - **Partial Optimization:** If some rules fail, successfully validated optimizations are retained while problematic ones are rejected
> - **Rule Confidence Scoring:** Low-confidence optimizations are flagged for manual review
>
> **3. Contradiction Resolution:**
> - **Priority-Based Resolution:** Critical constraints (timing, functionality) override optimization goals when conflicts arise
> - **Multi-Objective Balancing:** The elbow method (Equation 1) prevents selection of conflicting rules that would violate hardware constraints
> - **Verification Gate:** Two-tier verification (testbench + formal) catches constraint violations before final code generation
>
> **Edge Case Examples:**
> - **Memory overflow:** Detect when resource sharing would exceed available memory → revert to individual allocation
> - **Timing violations:** FSM state merging creating critical paths → maintain original state structure
> - **Power budget exceeded:** Resource sharing increasing dynamic power → prioritize power-optimized individual units
>
> ---
>
> **Summary:** We appreciate the reviewer's positive evaluation and constructive feedback. Our responses address the clarity concerns, provide the missing limitations discussion, and demonstrate SymRTLO's robustness across different domains and edge cases. We believe these clarifications further strengthen the paper's technical merit and practical impact.

---

> > ### Comment · Reviewer_u5E9 · 2025-08-02
> > **Rebuttal reviewer response**
> >
> > The authors have provided satisfactory clarifications and demonstrations, and the research manuscript presents significant technical merit. I recommend its acceptance.

---

> > > ### Author Response · Authors · 2025-08-03
> > >
> > > Dear Reviewer,
> > >
> > > Thank you for your thorough and constructive evaluation of our SymRTLO framework. We appreciate your clear summary of our contributions and your recognition of our background discussion, methodology, and experimental validation.
> > >
> > > Your suggestions for improving the ablation study’s clarity, explicitly addressing the framework’s limitations, and correcting minor typos are invaluable. We will revise Figure 5 and its caption for precision, add a dedicated “Limitations” section, and perform a comprehensive proofreading pass. We also welcome your questions about generalization across ASIC and FPGA designs, rule transferability across domains, and handling edge cases; we will address these points in detail in the revised manuscript.
> > >
> > > Thank you again for your insightful feedback and your recommendation for acceptance!

---

### Note · Authors · 2025-08-12

We thank all reviewers for their thoughtful engagement that strengthened our work and we thank the AC for supporting the review process. We're pleased that all reviewers found we addressed most of their concerns.

SymRTLO is a neuro-symbolic RTL optimizer that fuses LLMs, RAG rules, symbolic modules, and verification to align high-level intent with correct RTL. The consistent PPA gains over using synthesis tools alone make it a valuable addition to the EDA flow.

**Reviewer u5E9** found challenges, methodology, experiments, and validation clearly discussed, but asked about generalization and edge cases. Our rebuttal offered detailed explanations with examples. The reviewer wrote, **“The authors have provided satisfactory clarifications and demonstrations, and the research manuscript presents significant technical merit. I recommend its acceptance.”**

**Reviewer Ppek** recognized SymRTLO as a comprehensive framework and has questions for benchmark size, reasoning-model performance, and requested details on conflict resolution, RAG vs. search, and dataflow–control integration. We answered all 10 questions with new experiments; the reviewer found it **“answered most of my questions.”**

**Reviewer xZfF** highlights our open-source commitment, significant PPA gains, "simple yet effective" RAG+AST approach, and novel LLM-FSM integration. Initial concerns included limitations discussion, baseline comparisons, RAG implementation details, FSM scalability, more commercial tool comparisons, and checklist alignment.  After our comprehensive responses, the reviewer found their **concerns mostly addressed** and our **"comparative analysis … particularly impressive."**

**Reviewer b1CD** recognized our integration of LLMs as promising to explore, but questioned whether the optimizations required LLMs, raised reliability concerns, and requested more RAG examples. After our responses with concrete examples and testbench coverage experimental results, they acknowledged we **"addressed most concerns"** but requested **more evidence of unique capabilities over conventional synthesis tools.**

We clarify that: 1) SymRTLO complements, not replaces, traditional synthesis as a pre-synthesis optimizer, 2) Conventional synthesis tools have inherent limitations that only code-level optimization can overcome.

We provided extensive experimental results across 3 synthesis optimization levels and 2 detailed code examples demonstrating optimizations unachievable by synthesis alone.

---

### Decision · Program_Chairs · 2025-09-17

**Decision:**

Accept (poster)

**Comment:**

The paper introduces SymRTLO, a neuro-symbolic framework for RTL code optimization.

Strengths include:

- Clear discussion of challenges, methodology, and experiments; framework is well-articulated (u5E9).
- Promising direction of bringing LLM, RAG, and symbolic reasoning into compilation (Ppek, b1CD, xZfF).
- Significant PPA gains (xZfF).

Weaknesses include:
- Small benchmark size, limited evidence on scalability to large designs (Ppek).
- Missing comparisons with reasoning models and other LLM-based RTL generators (Ppek, xZfF).
- Some RAG rules and optimizations are trivial or standard in compilers, raising novelty concerns (xZfF, b1CD).
- Lacks evaluation with newer commercial tools (xZfF).
- LLM reliability and verification robustness (b1CD).

After the discussion, all reviewers agree to accept this paper.